# Task-Free Continual Learning via Online Discrepancy Distance Learning

**Fei Ye and Adrian G. Bors**

Department of Computer Science
University of York
York, YO10 5GH, UK
{fy689,adrian.bors}@york.ac.uk

## Abstract

Learning from non-stationary data streams, also called Task-Free Continual Learning (TFCL) remains challenging due to the absence of explicit task information in most applications. Even though recently some algorithms have been proposed for TFCL, these methods lack theoretical guarantees. Moreover, there are no theoretical studies about forgetting during TFCL. This paper develops a new theoretical analysis framework that derives generalization bounds based on the discrepancy distance between the visited samples and the entire information made available for training the model. This analysis provides new insights into the forgetting behaviour in classification tasks. Inspired by this theoretical model, we propose a new approach enabled with the dynamic component expansion mechanism for a mixture model, namely Online Discrepancy Distance Learning (ODDL). ODDL estimates the discrepancy between the current memory and the already accumulated knowledge as an expansion signal aiming to ensure a compact network architecture with optimal performance. We then propose a new sample selection approach that selectively stores the samples into the memory buffer through the discrepancy-based measure, further improving the performance. We perform several TFCL experiments with the proposed methodology, which demonstrate that the proposed approach achieves the state of the art performance.

## 1 Introduction

Continual learning (CL) and its extension to lifelong learning, represents one of the most desired functions in an artificial intelligence system, representing the capability of learning new concepts while maintaining the knowledge of past experiences [32]. Such an ability can be used in many real-time applications such as robotics, health investigative systems, autonomous vehicles [20] or for agents exploring artificial (meta) universes. Such applications require to continuously adapt to a changing environment. Modern deep learning models suffer from degenerated performances on past data after learning novel knowledge, a phenomenon called catastrophic forgetting [13].

A popular attempt to relieve forgetting in CL is by employing a small memory buffer to preserve a few past samples and replay them when training on a new task [3, 6]. However, when there are restrictions on the available memory capacity, memory-based approaches would suffer from degenerated performance on past tasks, especially when aiming to learn an infinite number of tasks. Recently, the Dynamic Expansion Model (DEM) [51] has shown promising results in CL, aiming to guarantee optimal performance by preserving the previously learnt knowledge through the parameters of frozen components trained on past data, while adding a new component when learning a novel task. However, such approaches require knowing where and when the knowledge associated with a given task is changed, which is not always applicable in a real environment.

36th Conference on Neural Information Processing Systems (NeurIPS 2022).

In this paper, we address a more realistic scenario, called Task-Free Continual Learning (TFCL) [2], where task identities are not available while the model can only access a small batch of samples at a given time. Most existing CL methods may be adapted to TFCL by removing the task information dependency. For instance, memory-based approaches can store a few past samples from the data stream at each training time and replay them later during the training [8, 12]. However, such an approach requires to carefully design the sample selection criterion to avoid memory overload. The key challenge for the memory-based approaches is the negative backward transfer caused by the stored samples which interferes with the process of model's updating with incoming samples [6]. This issue can be relieved by employing a DEM in which previously learnt samples are preserved into frozen components and would not interfere with the learning of new probabilistic data representations [24, 38]. However, these approaches do not provide any theoretical guarantees and there are no studies analysing the trade-off between the model's generalization and its complexity under TFCL.

There are some recent CL theoretical analysis studies from different perspectives including for the risk bounds [46, 51], NP-hard problems [17], Teacher-Student frameworks [23, 57] or for the game theory [37] applications. However all these approaches require strong assumptions, such as defining task identities, which are not available under the TFCL. This inspires us to bridge the gaps between the underlying theory and the algorithmic implementation for TFCL. In this study we propose a theoretical classification framework, which provides new insights in the forgetting behaviour analysis and guidance for algorithm design addressing catastrophic forgetting. The primary motivation behind the proposed theoretical framework is that we can formulate forgetting as a generalization error in the domain adaptation theory. Based on this analysis we extend the domain adaptation theory [29] to derive time-dependent generalization risk bounds, explicitly explaining the forgetting process at each training step. Inspired by the theory, we devise the Online Discrepancy Distance Learning (ODDL) method which introduces a new expansion mechanism based on the discrepancy distance estimation for implementing TFCL. The proposed expansion mechanism detects the data distribution shift by evaluating the variance of the discrepancy distance during the training. This model enables a trade-off mechanism between the model's generalization and complexity. We also propose a new sample selection approach based on the discrepancy-based criterion, which guides storing diverse samples with respect to the already learnt knowledge, further improving performance. Our contributions are :

- This paper is the first research study to propose a new theoretical framework for TFCL, which provides new insights into the forgetting behaviour of the model in classification tasks.
- Inspired by the theoretical analysis, we develop a novel dynamic expansion approach, which ensures a compact model architecture enabled by optimal performance.
- We propose a new sample selection approach that selects diverse data samples, with respect to the existing knowledge of the model, for the memory buffer, further improving performance.
- The proposed method achieves state of the art results on TFCL benchmarks,

## 2 Related works

**Continual learning** defines a learning paradigm which aims to learn a sequence of tasks without forgetting. Catastrophic forgetting is a major challenge in continual learning. One of the most popular approaches to relieve forgetting is by imposing a regularization loss within the optimization procedure [7, 11, 13, 16, 19, 25, 26, 31, 34, 35, 40, 41, 56], where the network's parameters deemed important to the past tasks are penalized when updating. Another kind of approaches for continual learning focuses on the memory system, which usually employs a small memory buffer [3, 5, 6, 28, 36, 44, 59] for storing some past data or trains a generator to provide the replay samples when learning new tasks [38, 43, 46, 47, 58, 57, 52]. However, these approaches usually rely on knowing the task information, which does not correspond to the context from the TFCL.

**Task-free continual learning** is a special scenario in CL where a model can only see one or very few samples in each training step/time without having any task labels. Using a small memory buffer to store past samples has shown benefits for TFCL and was firstly investigated in [2, 53, 55]. These memory replay approaches were then extended by employing Generative Replay Mechanisms (GRMs) for training both a Variational Autoencoder (VAEs) [15] and a classifier, where a new retrieving mechanism called the Maximal Interfered Retrieval (MIR) [1], is used to select specific data samples. The Gradient Sample Selection (GSS) [3] is another sample selection approach that treats sample selection as a constrained optimization reduction. More recently, a Learner-Evaluator

framework proposed for TFCL, called the Continual Prototype Evolution (CoPE) [8], stores the same number of samples for each class in the memory in order to ensure the balance replay. Another direction for the memory-based approaches, called the Gradient based Memory EDiting (GMED) [12], consists in editing the stored samples aiming to increase the loss in the upcoming model updates. This approach can be employed in existing CL models to further enhance their performance.

**Dynamic expansion models** aim to automatically increase the model's capacity to adapt to new tasks by adding new hidden layers and units. The Continual Unsupervised Representation Learning (CURL) [38], dynamically builds new inference models whenever meeting the expansion criterion. However, since CURL still requires a GRM to relieve forgetting, it would lead to a negative knowledge transfer when updating the network's parameters to adapt to a new task. This issue can be addressed by using Dirichlet processes by adding new components while freezing all previously learnt units, in a model called the Continual Neural Dirichlet Process Mixture (CN-DPM), [24]. However, the expansion criterion used by these approaches relies on changing the loss when training each time, which does not have any theoretical guarantees.

# 3 Theoretical analysis of TFCL

In this section, we firstly introduce the framework of learning settings and notations, and we analyze the forgetting behaviour for a single model, and then extend this for a dynamic expansion model by deriving their Generalization Bounds (GBs).

## 3.1 Preliminaries

Let $\mathcal{X}$ be the input space and $\mathcal{Y}$ represent the output space which is $\{-1, 1\}$ for binary classification and $\{1, 2, \ldots, n'\}, n' > 2$ for multi-class classification. Let $\mathcal{D}_i^T = \{\mathbf{x}_j^T, y_j^T\}_{j=1}^{N_i^T}$ and $\mathcal{D}_i^S = \{\mathbf{x}_j^S, y_j^S\}_{j=1}^{N_i^S}$ represent the training and testing sets for the $i$-th dataset where $\mathbf{x}_j^T \in \mathcal{X}$ and $y_j^T \in \mathcal{Y}$ are the image and the associated ground truth label. $N_i^T$ and $N_i^S$ represent the number of samples for $\mathcal{D}_i^T$ and $\mathcal{D}_i^S$, respectively. In this paper, we mainly focus on the task-free class-incremental learning, as described in the following.

**Definition 1. (Data stream.)** For a given $t$-th training dataset $\mathcal{D}_t^S$ with $C_t^S$ data categories, let us consider a data stream $\mathcal{S}$ consisting of samples $\mathcal{D}_{t,j}^S$ from each category, expressed as $S = \bigcup_{j=1}^{C_t^S} \mathcal{D}_{t,j}^S$. Let $\mathcal{D}_{t,j}^T$ represent the set of samples drawn from the $j$-th category of $\mathcal{D}_t^T$. Let $\mathbb{P}_{t,j}^T$ and $\mathbb{P}_{t,j}^S$ represent the distributions for $\mathcal{D}_{t,j}^T$ and $\mathcal{D}_{t,j}^S$, respectively. Let $\mathbb{P}_{t,j}^{T,\mathcal{X}}$ represent the marginal distribution over $\mathcal{X}$.

**Definition 2. (Learning setting.)** Let $\mathcal{T}_i$ represent the $i$-th training step. We assume that there are a total of $n$ training steps for learning $\mathcal{S}$, where during each training step $\mathcal{T}_i$ is learnt a small batch of paired samples $\{\mathbf{X}_i^b, \mathbf{Y}_i^b\}$, drawn from $\mathcal{S}$. We have $\mathcal{S} = \bigcup_{i=1}^n \{\mathbf{X}_i^b, \mathbf{Y}_i^b\}$, $\{\mathbf{X}_i^b, \mathbf{Y}_i^b\} \cap \{\mathbf{X}_j^b, \mathbf{Y}_j^b\} = \varnothing$, where $i \neq j$, and a model (classifier) can only access $\{\mathbf{X}_i^b, \mathbf{Y}_i^b\}$ at $\mathcal{T}_i$, while all previous batches are not available. After finishing all training steps, we evaluate the classification accuracy of the model on the testing set $\mathcal{D}_t^T$.

**Definition 3. (Model and memory.)** Let us consider a model $h$ implemented by a classifier, and $\mathcal{H} = \{h \mid h \colon \mathcal{X} \to \mathcal{Y}\}$ the space of classifiers. Let us consider a memory buffer $\mathcal{M}_i$, defined at $\mathcal{T}_i$, which continually adds new data to its memory at each training step and $\mathbb{P}_{\mathcal{M}_i}$ the probabilistic representation of $\mathcal{M}_i$. We also randomly remove samples from the memory buffer $\mathcal{M}_i$, in order to keep fixed its cardinality (number of samples), $|\mathcal{M}_i|$.

## 3.2 Measuring the distribution shift

In TFCL, the probabilistic distance between the target domain (testing set) and the source domain (memory buffer) would be dynamically changed during each training step. We can use the discrepancy distance [29] to measure this gap through analyzing the model's risk.

**Definition 4. (The risk.)** For a given distribution $\mathbb{P}_{t,j}^S$, the risk of a model $h$ is defined as $\mathcal{R}\left(h, \mathbb{P}_{t,j}^S\right) \triangleq \mathbb{E}_{\{\mathbf{x}, y\} \sim \mathbb{P}_{t,j}^S}\left[\mathcal{L}\left(y, h(\mathbf{x})\right)\right]$, where $\mathcal{L} \colon \mathcal{Y} \times \mathcal{Y} \to [0, 1]$ is the loss function.

**Definition 5. (Discrepancy distance.)** For two given distributions $\mathbb{P}_{t,j}^T$ and $\mathbb{P}_{t,j}^S$, the discrepancy distance $\mathcal{L}_d$ is defined on two marginals as :

$$\mathcal{L}_d\big(\mathbb{P}_{t,j}^{T,\mathcal{X}}, \mathbb{P}_{t,j}^{S,\mathcal{X}}\big) \triangleq \sup_{(h,h')\in\mathcal{H}^2} \left| \mathbb{E}_{\mathbf{x}\sim\mathbb{P}_{t,j}^{T,\mathcal{X}}}\big[\mathcal{L}\big(h(\mathbf{x}), h'(\mathbf{x})\big)\big] - \mathbb{E}_{\mathbf{x}\sim\mathbb{P}_{t,j}^{S,\mathcal{X}}}\big[\mathcal{L}\big(h(\mathbf{x}), h'(\mathbf{x})\big)\big] \right|, \quad (1)$$

where $\{h, h'\} \in \mathcal{H}$. In practice, the discrepancy distance $\mathcal{L}_d(\cdot, \cdot)$ can be estimated as the upper bound considering the Rademacher complexity which is used in the domain adaptation theory as a measure of richness for a particular hypothesis space [30, 60].

**Corollary 1.** For two given domains $\mathbb{P}_{t,j}^{T,\mathcal{X}}$ and $\mathbb{P}_{t,j}^{S,\mathcal{X}}$, let $U_\mathcal{P}$ and $U_\mathbb{P}$ represent sample sets of sizes $m_\mathcal{P}$ and $m_\mathbb{P}$, drawn independently from $\mathbb{P}_{t,j}^{T,\mathcal{X}}$ and $\mathbb{P}_{t,j}^{S,\mathcal{X}}$. Let $\widehat{\mathbb{P}}_{t,j}^{T,\mathcal{X}}$ and $\widehat{\mathbb{P}}_{t,j}^{S,\mathcal{X}}$ represent the empirical distributions for $U_\mathcal{P}$ and $U_\mathbb{P}$. Let $\mathcal{L}(\mathbf{x}', \mathbf{x}) = |\mathbf{x}' - \mathbf{x}|^1$ be a loss function (L1-Norm), satisfying $\forall(\mathbf{x}, \mathbf{x}') \in \mathcal{X}, \mathcal{L}(\mathbf{x}, \mathbf{x}') > M$, where $M > 0$. Then, with the probability $1 - \delta$, we have, [29] :

$$\mathcal{L}_d(\mathbb{P}_{t,j}^{T,\mathcal{X}}, \mathbb{P}_{t,j}^{S,\mathcal{X}}) \leq \mathcal{L}_d(\widehat{\mathbb{P}}_{t,j}^{T,\mathcal{X}}, \widehat{\mathbb{P}}_{t,j}^{S,\mathcal{X}}) + C^\star + 3M \left( \sqrt{\frac{\log\left(\frac{4}{\delta}\right)}{2m_\mathcal{P}}} + \sqrt{\frac{\log\left(\frac{4}{\delta}\right)}{2m_\mathbb{P}}} \right), \quad (2)$$

where $C^\star = 4q\left(\mathrm{Re}_{U_\mathcal{P}}(\mathcal{H}) + \mathrm{Re}_{U_\mathbb{P}}(\mathcal{H})\right)$ and $\mathrm{Re}_{U_\mathcal{P}}(\mathcal{H})$ is the Rademacher complexity (Appendix-B from Supplemental Material (SM)). Let $\mathcal{L}_{\widehat{d}}(\mathbb{P}_{t,j}^{T,\mathcal{X}}, \mathbb{P}_{t,j}^{S,\mathcal{X}})$ represent the Right-Hand Side (RHS) of Eq. (2). We also assume that $\mathcal{L}: \mathcal{Y} \times \mathcal{Y} \to [0, 1]$ is a symmetric and bounded loss function $\forall(y, y') \in \mathcal{Y}^2, \mathcal{L}(y, y') \leq U'$, and $\mathcal{L}(\cdot, \cdot)$ obeys the triangle inequality, where $U'$ is a positive number.

### 3.3 GB for a single model

Based on the definitions from Section 3.2, we firstly derive the GB for the learning process of a single model under TFCL.

**Theorem 1.** Let $\mathcal{P}_i$ represent the distribution of all visited training samples (including all previous batches) drawn from $\mathcal{S}$, at $\mathcal{T}_i$. Let $h_{\mathcal{P}_i} = \arg\min_{h\in\mathcal{H}} \mathcal{R}(h, \mathcal{P}_i)$ and $h_{\mathcal{M}_i} = \arg\min_{h\in\mathcal{H}} \mathcal{R}(h, \mathbb{P}_{\mathcal{M}_i})$ represent the ideal classifiers for $\mathcal{P}_i$ and $\mathbb{P}_{\mathcal{M}_i}$, respectively. We derive the GB between $\mathcal{P}_i$ and $\mathbb{P}_{\mathcal{M}_i}$, based on the results from Corollary 1 :

$$\mathcal{R}\big(h, \mathcal{P}_i\big) \leq \mathcal{R}\big(h, h_{\mathcal{M}_i}, \mathbb{P}_{\mathcal{M}_i}\big) + \mathcal{L}_{\widehat{d}}\big(\mathcal{P}_i^{\mathcal{X}}, \mathbb{P}_{\mathcal{M}_i}^{\mathcal{X}}\big) + \eta\big(\mathcal{P}_i, \mathbb{P}_{\mathcal{M}_i}\big), \quad (3)$$

where $\eta\big(\mathcal{P}_i, \mathbb{P}_{\mathcal{M}_i}\big)$ is the optimal combined error $\mathcal{R}(h_{\mathcal{P}_i}, h_{\mathcal{M}_i}, \mathcal{P}_i) + \mathcal{R}(h_{\mathcal{P}_i}, h_{\mathcal{P}_i}^\star, \mathcal{P}_i)$ where $\mathcal{R}(h_{\mathcal{P}_i}, h_{\mathcal{M}_i}, \mathbb{P}_{\mathcal{M}_i})$ is the risk, expressed by $\mathbb{E}_{\mathbf{x}\sim\mathbb{P}_{\mathcal{M}_i}}[\mathcal{L}(h_{\mathcal{P}_i}(\mathbf{x}), h_{\mathcal{M}_i}(\mathbf{x}))]$ and $h_{\mathcal{P}_i}^\star$ is the true labeling function for $\mathcal{P}_i$.

The proof is provided in Appendix-A from the SM. Compared to the GB used in the domain adaptation [29], Theorem 1 provides an explicit way to measure the gap between the model's predictions and the true labels in each training step ($\mathcal{T}_i$). During the initial training stages ($i$ is very small), the memory $\mathcal{M}_i$ can store all previous samples and GB is tight. However, as the number of training steps increases, the discrepancy distance $\mathcal{L}_{\widehat{d}}(\mathcal{P}_i^{\mathcal{X}}, \mathbb{P}_{\mathcal{M}_i}^{\mathcal{X}})$ would increase because $\mathcal{M}_i$ loses the knowledge corresponding to previously learnt data. This can lead to a degenerated performance on $\mathcal{P}_i$, corresponding to the forgetting process. Next we extend Theorem 1 to analyze the generalization performance on testing sets.

**Theorem 2.** For a given target domain $\mathbb{P}_{t,j}^T$, we derive the GB for a model at the training step $\mathcal{T}_i$ :

$$\mathcal{R}\big(h, \mathbb{P}_{t,j}^T\big) \leq \mathcal{R}\big(h, h_{\mathcal{M}_i}, \mathbb{P}_{\mathcal{M}_i}\big) + \mathcal{L}_{\widehat{d}}\big(\mathbb{P}_{t,j}^{T,\mathcal{X}}, \mathbb{P}_{\mathcal{M}_i}^{\mathcal{X}}\big) + \eta\big(\mathbb{P}_{t,j}^T, \mathbb{P}_{\mathcal{M}_i}\big), \quad (4)$$

The proof is similar to that for Theorem 1. We can observe that the generalization performance on a target domain $\mathbb{P}_{t,j}^T$, by a model $h$ is relying mainly on the discrepancy distance between $\mathbb{P}_{t,j}^{T,\mathcal{X}}$ and $\mathbb{P}_{\mathcal{M}_i}$. In practice, we usually measure the generalization performance of $h$ on several data categories where each category is represented by a different underlying distribution. In the following, we extend Theorem 2 for multiple target distributions.

**Lemma 1.** For a given data stream $S = \bigcup_{j=1}^{C_t^S} \mathcal{D}_{t,j}^S$ consisting of samples from $\mathcal{D}_t^S$, let $\mathcal{D}_t^T$ be the corresponding testing set and $\mathbb{P}_{t,j}^T$ represent the distribution of samples for the $j$-th category from

$\mathcal{D}_t^T$, we derive the GB for multiple target domains as:

$$\sum\nolimits_{j=1}^{C_t^T} \left\{ \mathcal{R}\big(h, \mathbb{P}_{t,j}^T\big) \right\} \leq \sum\nolimits_{j=1}^{C_t^T} \left\{ \mathcal{R}\big(h, h_{\mathcal{M}_i}, \mathbb{P}_{\mathcal{M}_i}\big) + \mathcal{L}_{\widehat{d}}\big(\mathbb{P}_{t,j}^{T,\mathcal{X}}, \mathbb{P}_{\mathcal{M}_i}^{\mathcal{X}}\big) + \eta\big(\mathbb{P}_{t,j}^T, \mathbb{P}_{\mathcal{M}_i}\big) \right\}, \quad (5)$$

where $C_t^T$ represents the number of testing data streams.

**Remarks.** Observations for Lemma 1 : 1) The optimal performance of the model $h$ on the testing set can be achieved by minimizing the discrepancy distance between each target domain $\mathbb{P}_{t,j}^{T,\mathcal{X}}$ and the distribution $\mathbb{P}_{\mathcal{M}_i}$ at the training step $\mathcal{T}_i$. 2) The study from [17] employs the set theory to theoretically demonstrate that a perfect memory is crucial for CL. In contrast, we evaluate the memory quality using the discrepancy distance in Eq. (5), which provides a practical way to investigate the relationship between the memory and forgetting behaviour of existing approaches [6, 8] at each training step without requiring any task information (see the Appendix-D from SM). 3) The study from [51] introduces a similar risk bound for forgetting analysis which requires the task information. In contrast, the proposed GB can be used in a more realistic CL scenario and we also provide the theoretical analysis for component diversity (Appendix-C from SM), which is missing in [51].

### 3.4 GB for the dynamic expansion mechanism

As discussed in Lemma 1, a memory of fixed capacity would lead to degenerated performance on all target domains. The other problem for a single memory system is the negative backward transfer [27] in which the performance of the model is decreased due to samples being drawn from entirely different distributions [14]. A Dynamic Expansion Model (DEM) can address these limitations from two aspects :1) DEM relieves the negative transfer by preserving the previously learnt knowledge into frozen components which are make up a mixture system; 2) DEM can achieve a better generalization performance under TFCL by allowing each component to model one or only a few similar underlying data distributions. We derive GB for DEM and show the advantages of a DEM for TFCL.

**Definition 7. (Dynamic expansion mechanism.)** Let $\mathcal{G}$ represent a dynamic expansion model, with $G_j$ the $j$-th component in $\mathcal{G}$, implemented by a single classifier. During the initial training phase, $\mathcal{G}$ starts with training its first component, and then adds new components during the following training steps. In order to overcome forgetting, only the newly created component is updated each time, while all previously trained components have their parameters frozen.

**Theorem 3.** For a given data stream $\mathcal{S} = \{\mathbf{X}_1^b, \cdots, \mathbf{X}_n^b\}$, let $\mathcal{P}_{(i,j)}$ represent the distribution of the $j$-th training batch $\mathbf{X}_j^b$ drawn from $\mathcal{S}$ (visited), at $\mathcal{T}_i$. We assume that $\mathcal{G} = \{G_1, \cdots, G_c\}$ trained $c$ components at $\mathcal{T}_i$. Let $\mathcal{T} = \{\mathcal{T}_{k_1}, \cdots, \mathcal{T}_{k_c}\}$ be a set of training steps, where $G_v$ was frozen at $\mathcal{T}_{k_v}$. We derive the GB for $\mathcal{G}$ at $\mathcal{T}_i$ as:

$$\frac{1}{i} \sum\nolimits_{j=1}^{i} \left\{ \mathcal{R}\big(h, \mathcal{P}_{(i,j)}\big) \right\} \leq \frac{1}{i} \sum\nolimits_{j=1}^{i} \left\{ \mathrm{F}_S\big(\mathcal{P}_{(i,j)}, \mathcal{G}\big) \right\}, \quad (6)$$

where $\mathrm{F}_S(\cdot, \cdot)$ is the selection function, defined as :

$$\mathrm{F}_S\big(\mathcal{P}_{(i,j)}, \mathcal{G}\big) \triangleq \min_{v=1,\cdots,c} \left\{ \mathcal{R}\big(h, h_{\mathcal{M}_{k_v}}, \mathbb{P}_{\mathcal{M}_{k_v}}\big) + \mathcal{L}_{\widehat{d}}\big(\mathcal{P}_{(i,j)}^{\mathcal{X}}, \mathbb{P}_{\mathcal{M}_{k_v}}^{\mathcal{X}}\big) + \eta\big(\mathcal{P}_{(i,j)}, \mathbb{P}_{\mathcal{M}_{k_v}}\big) \right\}, \quad (7)$$

where $\mathbb{P}_{\mathcal{M}_{k_v}}$ represents the memory distribution at $\mathcal{T}_{k_v}$. The proof is provided in Appendix-B from SM. Notice that $\mathrm{F}_S(\cdot, \cdot)$ can be used for arbitrary distributions. We assume an ideal model selection in Eq. (7), where always the component with the minimal risk is chosen. DEM can achieve the minimal upper bound to the risk (Left Hand Side (LHS) of Eq. (6)) when comparing with a single model (Theorem 3). Then, in the following we derive the GB for analyzing the generalization performance of $\mathcal{G}$ on multiple target distributions.

**Lemma 2.** For a given data stream $S = \bigcup_j^{C_t^S} \mathcal{D}_{t,j}^S$ consisting of samples from $\mathcal{D}_t^S$, we have a set of target sets $\{\mathcal{D}_{t,1}^T, \cdots, \mathcal{D}_{t,C_t^T}^T\}$, where each $\mathcal{D}_{t,j}^T$ contains $C_{t,j}^b$ batches of samples. Let $\mathbb{P}_{t,j}^T(d)$ represent the distribution of the $d$-th batch of samples in $\mathcal{D}_{t,j}^T$. We assume that $\mathcal{G}$ consists of $c$ components trained on samples from $\mathcal{S}$ at $\mathcal{T}_i$. We derive the GB for multiple target domains as :

$$\sum\nolimits_{j=1}^{C_t^T} \left\{ \sum\nolimits_{d=1}^{C_{t,j}^b} \mathcal{R}\big(h, \mathbb{P}_{t,j}^T(d)\big) \right\} \leq \sum\nolimits_{j=1}^{C_t^T} \left\{ \sum\nolimits_{d=1}^{C_{t,j}^b} \left\{ \mathrm{F}_S\big(\mathbb{P}_{t,j}^T(d), \mathcal{G}\big) \right\} \right\}. \quad (8)$$

**Remark.** We have several observations from Lemma 2 : 1) The generalization performance of $\mathcal{G}$ is relying on the discrepancy distance between each target distribution $\mathbb{P}_{t,j}^T$ and the memory distribution

$\mathbb{P}_{\mathcal{M}_{k_i}}$ of the selected component (more details in Appendix-C from SM). 2) Eq. (8) provides the analysis of the trade-off between the model's complexity and generalization for DEM [24, 38]. By adding new components, $\mathcal{G}$ would capture additional information of each target distribution and thus improve its performance. On the other hand, the selection process ensures a probabilistic diversity for the stored information, aiming to capture more knowledge with a minimal number of components.

In practice, we usually perform the model selection for $\mathcal{G}$ by using a certain criterion that only accesses the testing samples without task labels. Therefore, we introduce a selection criterion $\widehat{F}(\cdot, \cdot)$, implemented by comparing the sample log-likelihood :

$$\widehat{F}\big(\mathbb{P}_{t,j}^T(d), \mathcal{G}\big) \triangleq \operatorname*{arg\,max}_{v=1,\cdots,c} \big\{ \mathbb{E}_{\mathbf{x}\sim\mathbb{P}_{t,j}^T(d)}[\hat{f}(\mathbf{x}, G_{k_v})] \big\}, \tag{9}$$

where $\hat{f}(\cdot, \cdot)$ is a pre-defined sample-log likelihood function. Then Eq. (9) is used for model selection:

$$\widehat{F}_S\big(\mathbb{P}_{t,j}^T(d), \mathcal{G}\big) = \big\{ \mathcal{R}\big(h, h_{\mathcal{M}_s}, \mathbb{P}_{\mathcal{M}_s}\big) + \mathcal{L}_{\hat{d}}\big(\mathbb{P}_{t,j}^{T,\mathcal{X}}(d), \mathbb{P}_{\mathcal{M}_s}^{\mathcal{X}}\big) + \eta\big(\mathbb{P}_{t,j}^T(d), \mathbb{P}_{\mathcal{M}_s}\big) \mid$$
$$s = \widehat{F}\big(\mathbb{P}_{t,j}^T(d), \mathcal{G}\big) \big\}. \tag{10}$$

We rewrite Eq. (8) by using Eq. (10), resulting in :

$$\sum\nolimits_{j=1}^{C_t^T} \Big\{ \sum\nolimits_{d=1}^{C_{t,j}^b} \mathcal{R}\big(h, \mathbb{P}_{t,j}^T(d)\big) \Big\} \le \sum\nolimits_{j=1}^{C_t^T} \Big\{ \sum\nolimits_{d=1}^{C_{t,j}^b} \big\{ \widehat{F}_S\big(\mathbb{P}_{t,j}^T(d), \mathcal{G}\big) \big\} \Big\}. \tag{11}$$

Eq. (11), when compared with an ideal solution (Eq. (8)), would involve the extra error terms due to the selection process (Eq. (10)), expressed as $\sum_{j=1}^{C_t^T} \Big\{ \sum_{d=1}^{C_{t,j}^b} \big\{ \widehat{F}_S\big(\mathbb{P}_{t,j}^T(d), \mathcal{G}\big) - F_S\big(\mathbb{P}_{t,j}^T(d), \mathcal{G}\big) \big\} \Big\}$. In Section 4, we introduce a new CL framework according to the theoretical analysis.

## 3.5 Parallels with other studies defining lifelong learning bounds

In this section, we discuss the differences between the results in this paper and those from other studies proposing lifelong learning bounds. The bound from Theorem 1 in [30] assumes that the model is trained on all previous samples, which is not practical under a TFCL setting. In contrast, in Theorem 1 from this paper, the model is trained on a memory buffer, which can be used in the context of TFCL. In addition, the bounds from Theorem 1 [30] mainly provide the theoretical guarantees for the performance when learning a new data distribution, which does not represent an analysis of the model's forgetting. In contrast, this paper studies the forgetting analysis of our model and its theoretical developments can be easily extended for analyzing the forgetting behaviour of different continual learning methods, while this cannot be said about the study from [30] (See Appendix-D from supplemental material).

When comparing with [33], our theoretical analysis does not rely on explicit task boundaries, which is a more realistic assumption for TFCL. In addition, [33] employs the KL divergence to measure the distance between two distributions, which would require knowing the explicit distribution form and is thus hard to evaluate in practice during the learning. However, our theory employs the discrepancy distance, which can be reliably estimated. Moreover, the study from [33] does not develop a practical algorithm to be used according to the theoretical analysis. In contrast, our theory provides guidelines for the algorithm design under the TFCL assumption. Finally, inspired by the proposed theoretical analysis, this paper develops a successful continual learning approach for TFCL.

Theorem 1 in [4], similarly to the study from [30], assumes that the model is trained on uncertain data sets over time, which would not be suitable for TFCL since the model can not access previously learnt samples under TFCL. In contrast, Theorem 1 in our theory represents a realistic TFCL scenario in which the model is trained on a fixed-length memory buffer and is evaluated on all previously seen samples. Therefore, our analysis for the forgetting behaviour of the model under TFCL relies on Theorem 1 and its consequences. In addition, the study from [4], similarly to [30, 33], only provides a theoretical guarantee for a single model. In contrast, we extend our theoretical analysis to the dynamic expansion model, which motivates us to develop a novel continual learning approach for TFCL. Moreover, this paper is the first work to provide the theoretical analysis for the diversity of knowledge recorded by different components (See Appendix-C from supplemental material). This analysis indicates that by maintaining the knowledge diversity among different components we ensure a good trade-off between the model's complexity and the generalization performance, thus providing invaluable insights into algorithm design for TFCL.

## 4 Methodology

### 4.1 Network architecture

Each component $G_j \in \mathcal{G}$ consists of a classifier $h_j$ implemented by a neural network $f_{\varsigma_j}(\mathbf{x})$ with trainable parameters $\varsigma_j$, and a variational autoencoder (VAE) model implemented by an encoding network $\mathrm{enc}_{\omega_j}$ as well as the corresponding decoding network $\mathrm{dec}_{\theta_j}$, with trainable parameters $\{\omega_j, \theta_j\}$. Due to its robust generation and inference mechanisms, VAEs have been widely used in many applications [48, 49, 50, 51, 54, 61]. This paper employs a VAE for discrepancy estimation and component selection. The loss function for the $j$-th component at $\mathcal{T}_i$ is defined as :

$$\mathcal{L}_{class}(G_j, \mathcal{M}_i) \triangleq \frac{1}{|\mathcal{M}_i|} \sum\nolimits_{t=1}^{|\mathcal{M}_i|} \left\{ \mathcal{L}_{ce}\big(h_j(\mathbf{x}_t), y_t\big) \right\}, \tag{12}$$

$$\mathcal{L}_{VAE}(G_j, \mathcal{M}_i) \triangleq -\mathbb{E}_{q_{\omega_j}(\mathbf{z}\,|\,\mathbf{x})} \left[ \log p_{\theta_j}(\mathbf{x}_t\,|\,\mathbf{z}) \right] + D_{KL} \left[ q_{\omega_j}(\mathbf{z}\,|\,\mathbf{x}_t) \,||\, p(\mathbf{z}) \right], \tag{13}$$

where $\{\mathbf{x}_t, y_t\} \sim \mathcal{M}_i$, where $|\mathcal{M}_i|$ is the memory buffer size, and $\mathcal{L}_{ce}(\cdot)$ is the cross-entropy loss. $\mathcal{L}_{VAE}(\cdot, \cdot)$ is the VAE loss [15], $p_{\theta_j}(\mathbf{x}_t\,|\,\mathbf{z})$ and $q_{\omega_j}(\mathbf{z}\,|\,\mathbf{x}_t)$ are the encoding and decoding distributions, implemented by $\mathrm{enc}_{\omega_j}$ and $\mathrm{dec}_{\theta_j}$, respectively. We also implement $\hat{f}(\cdot, \cdot)$ in Eq. (9) by $-\mathcal{L}_{VAE}(\cdot, \cdot)$ for the component selection at the testing phase. The training algorithm for the proposed ODDL consists of three stages (See more information in Appendix-E from SM). In the initial training stage, we aim to learn the initial knowledge of the data stream and preserve it into a component $G_1$ with frozen parameters, which can provide information for the dynamic expansion and can be sampled for the selection evaluation in the subsequent learning. During the evaluator training stage we train the current component as the evaluator that estimates the discrepancy distance between each previously learnt component and the memory buffer, deciding the model expansion. In the sample selection stage, we train the current component with new data, while we aim to promote knowledge diversity among mixture's components.

In the following, we firstly propose a new dynamic expansion mechanism based on the discrepancy criterion. Then we introduce a new sample selection approach for the memory buffer further improving the performance of the model. Finally, we provide the detailed algorithm implementation.

### 4.2 Discrepancy based mixture model expansion

From Lemma 2, we observe that the probabilistic diversity of the trained components in $\mathcal{G}$ can ensure a compact network architecture while maintaining a good generalization performance (See Theorem 4 in Appendix-C from SM). In order to achieve this, we maximize the discrepancy between the current memory buffer and the probabilistic representation of each trained component of $\mathcal{G}$, during the training, by using the discrepancy distance (notations are defined in Theorem 3), expressed as :

$$\mathcal{M}^\star = \arg\max_{\mathcal{M}_i} \sum\nolimits_j^c \left\{ \mathcal{L}_{\widehat{d}}(\mathbb{P}_{\mathcal{M}_{k_j}}^{\mathcal{X}}, \mathbb{P}_{\mathcal{M}_i}^{\mathcal{X}}) \right\}, \tag{14}$$

where $i = k_c + 1, \cdots, n$ represents the index of the training steps and $k_c$ is the $c$-th component trained at $\mathcal{T}_{k_c}$. $\mathcal{M}^\star$ is the closest learnt distribution of one of the components to that of the memory buffer. Eq. (14) can be seen as a recursive optimization problem when $\mathcal{G}$ dynamically adds new components ($c$ is increased) during the training. We derive an expansion criterion, based on the discrepancy distance from Eq. (14), while balancing the model's complexity and performance, :

$$\min_{j=1}^c \left\{ \mathcal{L}_{\widehat{d}}(\mathbb{P}_{\mathcal{M}_{k_j}}^{\mathcal{X}}, \mathbb{P}_{\mathcal{M}_i}^{\mathcal{X}}) \right\} \geq \lambda, \tag{15}$$

where $\lambda \in [0, 4]$ is an architecture expansion threshold chosen empirically. If the current memory distribution $\mathbb{P}_{\mathcal{M}_i}$ is sufficiently different from each component (satisfies Eq. (15)), $\mathcal{G}$ will add a new component to preserve the knowledge of the current memory $\mathcal{M}_i$, while also encouraging the probabilistic diversity among the trained components.

In the following, we describe the implementation. We start by training the first component $G_1$ which consists of a classifier $h_1$ and a VAE model $v_1$, which is used for model selection at the testing phase. We also train an additional component called the evaluator $G_e = \{h_e, v_e\}$, at the initial training stage, which aims to capture the future information about the data stream. Once the memory $\mathcal{M}_j$ reaches its maximum size at $\mathcal{T}_j$, we freeze the weights of the first component to preserve the previously learnt

knowledge while continually training the evaluator during the following training steps. Since the evaluator continually captures the knowledge from the current memory $\mathcal{M}_i$, we check the model expansion using Eq. (15) at $\mathcal{T}_i$ :

$$\mathcal{L}_d^\star\big(\mathbb{P}^{\mathcal{X}}_{\mathcal{M}_{k_1}}, \mathbb{P}^{\mathcal{X}}_{\mathcal{M}_i}\big) \geq \lambda\,. \tag{16}$$

Notice that we can not access the previously learnt memory distributions $\mathbb{P}^{\mathcal{X}}_{\mathcal{M}_{k_1}}$ and we approximate them by using the auxiliary distributions $\mathbb{P}^{\mathcal{X}}_{v_1^j}$ formed by samples $v_1^j$ drawn from $G_1^j$, where the superscript $j$ represents the first component finishing the training at $\mathcal{T}_j$. $\mathcal{L}_d^\star(\cdot, \cdot)$ is the estimator of the discrepancy distance, achieved by $h_e^i$ and $h_1^j$ :

$$\mathcal{L}_d^\star\big(\mathbb{P}^{\mathcal{X}}_{v_1^j}, \mathbb{P}_{\mathcal{M}_i}\big) \triangleq \Big|\mathbb{E}_{\mathbf{x} \sim \mathbb{P}^{\mathcal{X}}_{v_1^j}}\big[\mathcal{L}\big(h_1^j(\mathbf{x}), h_e^i(\mathbf{x})\big)\big] - \mathbb{E}_{\mathbf{x} \sim \mathbb{P}_{\mathcal{M}_i}}\big[\mathcal{L}\big(h_1^j(\mathbf{x}), h_e^i(\mathbf{x})\big)\big]\Big|. \tag{17}$$

If Eq. (16) is fulfilled, then we add $G_e^i$ into the model $\mathcal{G}$, while building a new evaluator ($G_e^{i+1} = G_3^{i+1}$) at $\mathcal{T}_{i+1}$; otherwise, we train $G_e^i \to G_e^{i+1}$ at the next training step, $\mathcal{T}_{i+1}$. Furthermore, for satisfying the diversity of knowledge in the components from the mixture model (See Lemma 2), we clear the memory $\mathcal{M}_i$ when performing the expansion in order to ensure that we would learn non-overlapping distributions during the following training steps. As illustrated in Fig. 1 we can also extend Eq. (16) to the expansion criterion for $\mathcal{G} = \{G_1, \cdots, G_{s-1}\}$ after creating and freezing $(s-1)$ components :

$$\min_{j=1}^{s-1} \Big\{ \mathcal{L}_d^\star\big(\mathbb{P}^{\mathcal{X}}_{v_j^{k_j}}, \mathbb{P}^{\mathcal{X}}_{\mathcal{M}_i}\big)\Big\} \geq \lambda\,, \tag{18}$$

where $\mathbb{P}^{\mathcal{X}}_{v_j^{k_j}}$ is the distribution of the generated samples by $G_j^{k_j}$ and we denote $S_j = \mathcal{L}_d^\star\big(\mathbb{P}^{\mathcal{X}}_{v_j^{k_j}}, \mathbb{P}_{\mathcal{M}_i}\big)$.

### 4.3 Sample selection

According to Lemma 2, the probabilistic diversity of the knowledge accumulated in the components is crucial for the performance. In the following, we also introduce a novel sample selection approach for the memory buffer that further encourages the diversity of the learnt information. The primary motivation behind the proposed sample selection approach is that we desire to store in the memory buffer those samples that are completely different from the data used for training existing components. This mechanism enables the newly created component to capture a different underlying data distribution. To implement this goal, we estimate the discrepancy distance on a pair of samples as :

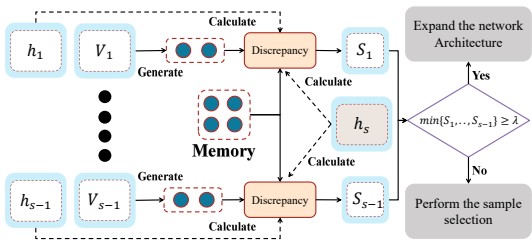

Figure 1: We generate the known information by using the VAE ($V_j$) of each previous component $G_j, j = 1, \cdots, s-1$, used to evaluate the discrepancy distance $S_j = \mathcal{L}_d^\star\big(\mathbb{P}^{\mathcal{X}}_{v_j}, \mathbb{P}_{\mathcal{M}_i}\big)$ at $\mathcal{T}_i$ (Eq. (17)), between $G_j$ and the memory buffer. Then we use these discrepancy scores $\{S_1, \cdots, S_{s-1}\}$ to check the model expansion (Eq. (18))

$$\mathcal{L}_d^s(\mathcal{G}, \mathbf{x}_c) \triangleq \frac{1}{s}\sum_{t=1}^{s} \big|\mathcal{L}(h_t^{k_t}(\mathbf{x}_c), h_e^i(\mathbf{x}_c)) - \mathcal{L}(h_t^{k_t}(\mathbf{x}_t^{k_t}), h_e^i(\mathbf{x}_t^{k_t}))\big|\,, \tag{19}$$

where $\mathbf{x}_c$ is the $c$-th sample from $\mathcal{M}_i$ while $\mathbf{x}_t^{k_t}$ is drawn from $G_t^{k_t}$. $h_t^{k_t}(\cdot)$ is the classifier of $G_t^{k_t}$ from $\mathcal{G}$. Eq. (19) evaluates the average discrepancy distance between the knowledge generated by each trained component and the stored samples, guiding the sample selection at the training step ($\mathcal{T}_i$) :

$$\mathcal{M}_i = \bigcup_{j=1}^{|\mathcal{M}_i'|-b} \mathcal{M}_i'[j]\,, \tag{20}$$

where $\mathcal{M}_i'$ is the sorted memory that satisfies the condition $\mathcal{L}_d^s(\mathcal{G}, \mathcal{M}_i'[a]) > \mathcal{L}_d^s(\mathcal{G}, \mathcal{M}_i'[q])$ for $a < q$. $\mathcal{M}_i'[j]$ represents the $j$-th stored sample and $d = 10$ is the batch size used in the experiments. We name the Online Discrepancy Distance Learning with sample Selection as ODDL-S.

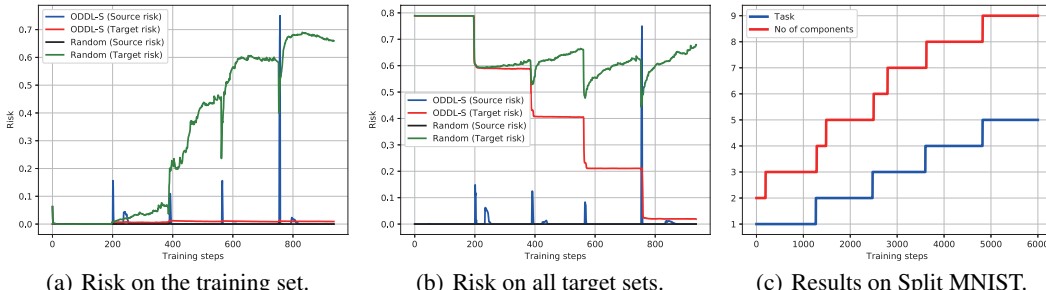

| (a) Risk on the training set. | (b) Risk on all target sets. | (c) Results on Split MNIST. |

Figure 2: The forgetting analysis of a single model and DEM in (a) and (b) where the batch size is 64 in the data stream. Data distribution shift and increasing the number of components in ODDL-S during the training in (c).

### 4.4 Algorithm implementation

The algorithm has three main stages (pseudocode is provided in **Algorithm 1**, Appendix-E from SM) :

- **(Initial training).** We start by building two components $\{G_1, G_2\}$ where we only add $G_1$ in $\mathcal{G}$ and consider $G_e = G_2$ as the Evaluator. $G_1$ and $G_2$ are trained jointly using Eq. (13) and Eq. (12) during the initial training stage until the memory $\mathcal{M}_j$ reaches its maximum size $|\mathcal{M}|^{max}$ at a certain training step $\mathcal{T}_j$. Then we freeze the first component $G_1$.
- **(Evaluator training).** In this stage, we only update the Evaluator using Eq. (13) and Eq. (12) on $\mathcal{M}_{j+1}$ at $\mathcal{T}_{j+1}$. If $|\mathcal{M}_{j+1}| \geq |\mathcal{M}|^{max}$, then we evaluate the discrepancy distance using Eq. (17) to check the expansion (Eq. (18)). If the expansion criterion is satisfied we add $G_e$ to $\mathcal{G}$ and build a new Evaluator while clearing up the memory $\mathcal{M}_{j+1}$, otherwise, we perform the sample selection.
- **(Sample selection).** We evaluate the diversity score for each stored sample in $\mathcal{M}_{j+1}$ using Eq. (19) and perform the sample selection for $\mathcal{M}_{j+1}$ using Eq. (20). Then we return to the second stage.

## 5 Experiments

We perform the experiments to address the following research questions: 1) What factors would cause the model's forgetting, and how to explain such behaviour? 2) How efficient is the proposed ODDL-S under TFCL benchmarks? 3) How important is each module in OODL-S?

In this experiment, we adapt the TFCL setting from [8] which employs several datasets including Split MNIST [22, 53], Split CIFAR10 [18, 53] and Split CIFAR100 [18]. The detailed information for datasets, hyperparameters and network architectures is provided in Appendix-F from the supplementary material. The code is available at `https://dtuzi123.github.io/ODDL/`.

### 5.1 Empirical results for the forgetting analysis

In this section, we investigate the forgetting behaviour of the proposed model according to the theoretical framework. Firstly, we train a single classifier $h$ on the Split MNIST database, as a baseline. We consider a memory buffer of maximum size of 2000, with the batch size of 10; the extra stored samples are randomly removed when the memory is full. We evaluate the source and target risks for the proposed ODDL under Split MNIST and plot the results in Fig. 2. We estimate the target risk on all visited training samples $\mathcal{R}(h, \mathcal{P}_i)$ and the source risk on the memory $\mathcal{R}(h, h_{\mathcal{M}_i}, \mathbb{P}_{\mathcal{M}_i})$. We plot the results in Fig. 2-a, where "Random (Source risk)" represents the source risk of a single classifier, considered as a baseline. At the same time, the baseline tends to increase the target risk on $\mathcal{P}_i$ with the training steps, as shown in Fig. 2-a. This happens because the memory loses the samples associated with the previous tasks, theoretically explained by Theorem

Table 1: The accuracy of various continual learning models for five independent runs.

| Methods | Split MNIST | Split CIFAR10 | Split CIFAR100 |
|---|---|---|---|
| finetune* | $19.75 \pm 0.05$ | $18.55 \pm 0.34$ | $3.53 \pm 0.04$ |
| GEM* | $93.25 \pm 0.36$ | $24.13 \pm 2.46$ | $11.12 \pm 2.48$ |
| iCARL* | $83.95 \pm 0.21$ | $37.32 \pm 2.66$ | $10.80 \pm 0.37$ |
| reservoir* | $92.16 \pm 0.75$ | $42.48 \pm 3.04$ | $19.57 \pm 1.79$ |
| MIR* | $93.20 \pm 0.36$ | $42.80 \pm 2.22$ | $20.00 \pm 0.57$ |
| GSS* | $92.47 \pm 0.92$ | $38.45 \pm 1.41$ | $13.10 \pm 0.94$ |
| CoPE-CE* | $91.77 \pm 0.87$ | $39.73 \pm 2.26$ | $18.33 \pm 1.52$ |
| CoPE* | $93.94 \pm 0.20$ | $48.92 \pm 1.32$ | $21.62 \pm 0.69$ |
| ER + GMED† | $82.67 \pm 1.90$ | $34.84 \pm 2.20$ | $20.93 \pm 1.60$ |
| ER$_a$ + GMED† | $82.21 \pm 2.90$ | $47.47 \pm 3.20$ | $19.60 \pm 1.50$ |
| CURL* | $92.59 \pm 0.66$ | - | - |
| CNDPM* | $93.23 \pm 0.09$ | $45.21 \pm 0.18$ | $20.10 \pm 0.12$ |
| Dynamic-OCM | $94.02 \pm 0.23$ | $49.16 \pm 1.52$ | $21.79 \pm 0.68$ |
| ODDL | $94.85 \pm 0.02$ | $51.48 \pm 0.12$ | $26.20 \pm 0.72$ |
| ODDL-S | $\mathbf{95.75 \pm 0.05}$ | $\mathbf{52.69 \pm 0.11}$ | $\mathbf{27.21 \pm 0.87}$ |

1. We also evaluate the risk of the baseline on all testing sets (target risk) and plot the results in Fig. 2-b, where we can observe that the baseline invariably leads to a large target risks even when the number of training steps increases. Meanwhile, the performance of ODDL on the distribution $\mathcal{P}_i$ (target risk) does not degenerate during the whole training phase. These results show that ODDL can relieve forgetting and achieve better generalization than the baseline on all target sets, while expanding its architecture to learn novel knowledge.

## 5.2 Results on TFCL benchmark

We provide the results in Tab. 1 where * and † denote the results cited from [8] and [12], respectively. We compare with several baselines including: finetune that directly trains a classifier on the data stream, GSS [3], Dynamic-OCM [53], MIR [1], Gradient Episodic Memory (GEM) [27], Incremental Classifier and Representation Learning (iCARL) [39], Reservoir [45], CURL, CNDPM, CoPE, ER + GMED and $ER_a$ + GMED [12] where ER is the Experience Replay [42] and $ER_a$ is ER with data augmentation. The number of components in ODDL-S and ODDL for Split MNIST, Split CIFAR10 and Split CIFAR100 is 7, 9, 7, respectively. From Tab. 1 the proposed ODDL-S outperforms CNDPM, which uses more parameters, on all datasets and achieves state-of-the-art performance.

In the following, we also evaluate the performance of the proposed approach on the large-scale dataset (MINI-ImageNet [21]), and Permuted MNIST [9]. We split MINI-ImageNet into 20 tasks (See details in Appendix-F1 from SM), namely Split MImageNet. We follow the setting from [1] where the maximum memory size is $10K$, and a smaller version of ResNet-18 [10] is used as the classifier. The hyperparameter $\lambda$ used in Eq. (15) for expanding ODDL when learning Split MINI-ImageNet and Permuted MNIST (where pixels are randomly premuted in images) is equal to 1.2 and 1.5, respectively. In Tab. 2 we compare with several state-of-the-art methods under Split MImageNet, where the results of other baselines are from [12]. From these results, ODDL-S outperforms different baselines under the challenging dataset.

Table 2: Classification accuracy for 20 runs when testing various models on Split MImageNet and Permuted MNIST.

| Methods | Split MImageNet | Permuted MNIST |
|---|---|---|
| $ER_a$ | $25.92 \pm 1.2$ | $78.11 \pm 0.7$ |
| ER + GMED | $27.27 \pm 1.8$ | $78.86 \pm 0.7$ |
| MIR+GMED | $26.50 \pm 1.3$ | $79.25 \pm 0.8$ |
| MIR | $25.21 \pm 2.2$ | $79.13 \pm 0.7$ |
| CNDPM | $27.12 \pm 1.5$ | $80.68 \pm 0.7$ |
| ODDL | $27.45 \pm 0.9$ | $82.33 \pm 0.6$ |
| ODDL-S | $\mathbf{28.68} \pm 1.5$ | $\mathbf{83.56} \pm 0.5$ |

## 5.3 Ablation study

We investigate the expansion of the ODDL-S according to the proposed discrepancy-based criterion. In Fig. 2-c we show the number of components in each training step for ODDL-S, when training on Split MNIST, where the original MNIST database is learnt segment by segment. In Fig. 2-c "Task" represents the number of tasks in each training step. We observe that the proposed discrepancy-based criterion can detect the data distribution shift accurately, allowing ODDL-S to expand the network architecture each time when detecting the data distribution shift. This also encourages the proposed ODDL-S to use a minimal number of components while achieving optimal performance, as discussed in Lemma 2. We also provide the analysis of how to maximize the memory bound in Appendix-F2.2, while more ablation study results are provided in Appendix-F.2 from supplementary material.

## 6 Conclusion

In this paper, we develop a novel theoretical framework for Task-Free Continual Learning (TFCL), by defining a statistical discrepancy distance between given data distributions and those learnt by the lifelong learning model. Inspired by the theoretical analysis, we propose the Online Discrepancy Distance Learning enabled by a selective memory buffer sampling approach (ODDL-S) model, which trades off between the model's complexity and performance. The memory buffer sampling mechanism ensures the information diversity learning. The proposed theoretical analysis provides new insights into the model's forgetting behaviour during each training step of TFCL. Experimental results on several TFCL benchmarks show that the proposed ODDL-S achieves state-of-the-art performance.

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
