# Appendix for Task-Free Continual Learning Via Online Discrepancy Distance Learning

**Fei Ye and Adrian G. Bors**
Department of Computer Science
University of York
York, YO10 5GH, UK
{fy689,adrian.bors}@york.ac.uk

## Contents

36th Conference on Neural Information Processing Systems (NeurIPS 2022).

# A The proof of Theorem 1

**Theorem 1.** Let $\mathcal{P}_i$ represent the distribution of all seen training samples (including all previous batches) drawn from $\mathcal{S}$ at $\mathcal{T}_i$. Let $h_{\mathcal{P}_i} = \arg\min_{h \in \mathcal{H}} \mathcal{R}(h, \mathcal{P}_i)$ and $h_{\mathcal{M}_i} = \arg\min_{h \in \mathcal{H}}(h, \mathbb{P}_{\mathcal{M}_i})$ represent the ideal classifiers for $\mathcal{P}_i$ and $\mathbb{P}_{\mathcal{M}_i}$, respectively. We derive a risk bound between $\mathcal{P}_i$ and $\mathbb{P}_{M_i}$ as :

$$\mathcal{R}(h, \mathcal{P}_i) \leq \mathcal{R}(h, h_{\mathcal{M}_i}, \mathbb{P}_{\mathcal{M}_i}) + \mathcal{L}_{\widehat{d}}(\mathcal{P}_i^{\mathcal{X}}, \mathbb{P}_{\mathcal{M}_i}^{\mathcal{X}}) + \eta(\mathcal{P}_i, \mathbb{P}_{\mathcal{M}_i}), \tag{1}$$

where $\eta(\cdot, \cdot)$ is the optimal combined error defined as :

$$\eta(\mathcal{P}_i, \mathbb{P}_{\mathcal{M}_i}) = \mathcal{R}(h_{\mathcal{P}_i}, h_{\mathcal{M}_i}, \mathcal{P}_i) + \mathcal{R}(h_{\mathcal{P}_i}, h_{\mathcal{P}_i}^{\star}, \mathcal{P}_i), \tag{2}$$

where $\mathcal{R}(h_{\mathcal{P}_i}, h_{\mathcal{M}_i}, \mathbb{P}_{\mathcal{M}_i}) = \mathbb{E}_{\mathbf{x} \sim \mathbb{P}_{\mathcal{M}_i}}[\mathcal{L}(h_{\mathcal{P}_i}(\mathbf{x}), h_{\mathcal{M}_i}(\mathbf{x}))]$ and $h_{\mathcal{P}_i}^{\star}$ is the true labeling function for $\mathcal{P}_i$.

**Proof.** Firstly, we consider $\mathcal{R}(h, \mathcal{P}_i) \equiv \mathcal{R}(h, h_{\mathcal{P}_i}^*, \mathcal{P}_i)$. We adopt similar derivations to those used for Domain Adaptation theory [12]. We consider fixing the classifier $h \in \mathcal{H}$. According to the triangle inequality property of $\mathcal{R}(\cdot, \cdot)$ and $\mathcal{R}(\cdot, \cdot, \cdot)$ and *Definition 5* from the paper, see the discrepancy distance $\mathcal{L}_d(\cdot, \cdot)$ from equation (2) from the paper, we have:

$$\begin{aligned}
\mathcal{R}(h, h_{\mathcal{P}_i}^{\star}, \mathcal{P}_i) &\leq \mathcal{R}(h, h_{\mathcal{M}_i}, \mathcal{P}_i) + \mathcal{R}(h_{\mathcal{M}_i}, h_{\mathcal{P}_i}, \mathcal{P}_i) + \mathcal{R}(h_{\mathcal{P}_i}^{\star}, h_{\mathcal{P}_i}, \mathcal{P}_i) \\
&\leq \mathcal{R}(h, h_{\mathcal{M}_i}, \mathbb{P}_{\mathcal{M}_i}) + \mathcal{R}(h_{\mathcal{M}_i}, h_{\mathcal{P}_i}, \mathcal{P}_i) + \mathcal{R}(h_{\mathcal{P}_i}, h_{\mathcal{P}_i}^{\star}, \mathcal{P}_i) + \mathcal{L}_d(\mathcal{P}_i^{\mathcal{X}}, \mathbb{P}_{\mathcal{M}_i}^{\mathcal{X}})
\end{aligned} \tag{3}$$

By considering the discrepancy distance (Definition 5 of the paper), we rewrite Eq. (3) as :

$$\mathcal{R}(h, \mathcal{P}_i) \leq \mathcal{R}(h, h_{\mathcal{M}_i}, \mathbb{P}_{\mathcal{M}_i}) + \mathcal{L}_d(\mathcal{P}_i^{\mathcal{X}}, \mathbb{P}_{\mathcal{M}_i}^{\mathcal{X}}) + \eta(\mathcal{P}_i, \mathbb{P}_{\mathcal{M}_i}), \tag{4}$$

In addition, from Corollary 1 of the paper, we have :

$$\begin{aligned}
\mathcal{L}_d(\mathcal{P}_i^{\mathcal{X}}, \mathbb{P}_{\mathcal{M}_i}^{\mathcal{X}}) \leq{} & \mathcal{L}_d(\widehat{\mathcal{P}}_i^{\mathcal{X}}, \widehat{\mathbb{P}}_{\mathcal{M}_i}^{\mathcal{X}}) \\
&+ 4q \left( \mathrm{Re}_{U_{\mathcal{P}_i^{\mathcal{X}}}}(\mathcal{H}) + \mathrm{Re}_{U_{\mathbb{P}_{\mathcal{M}_i}^{\mathcal{X}}}}(\mathcal{H}) \right) \\
&+ 3M \left( \sqrt{\frac{\log\left(\frac{4}{\delta}\right)}{2m_{\mathcal{P}_i^{\mathcal{X}}}}} + \sqrt{\frac{\log\left(\frac{4}{\delta}\right)}{2m_{\mathbb{P}_{\mathcal{M}_i}^{\mathcal{X}}}}} \right).
\end{aligned} \tag{5}$$

where $U_{\mathcal{P}_i^{\mathcal{X}}}$ and $U_{\mathbb{P}_{\mathcal{M}_i}^{\mathcal{X}}}$ represent samples of sizes $m_{\mathcal{P}_i^{\mathcal{X}}}$ and $m_{\mathbb{P}_{\mathcal{M}_i}^{\mathcal{X}}}$, drawn independently from $\mathcal{P}_i^{\mathcal{X}}$ and $\mathbb{P}_{\mathcal{M}_i}^{\mathcal{X}}$. $\widehat{\mathcal{P}}_i^{\mathcal{X}}$ and $\widehat{\mathbb{P}}_{\mathcal{M}_i}^{\mathcal{X}}$ represent the empirical distributions of $U_{\mathcal{P}_i^{\mathcal{X}}}$ and $U_{\mathbb{P}_{\mathcal{M}_i}^{\mathcal{X}}}$. Then we replace $\mathcal{L}_d(\mathcal{P}_i^{\mathcal{X}}, \mathbb{P}_{\mathcal{M}_i}^{\mathcal{X}})$ of Eq. (4) by the right-hand side of Eq. (5), resulting in :

$$\begin{aligned}
\mathcal{R}(h, \mathcal{P}_i) \leq{} & \mathcal{R}(h, h_{\mathcal{M}_i}, \mathbb{P}_{\mathcal{M}_i}) + \eta(\mathcal{P}_i, \mathbb{P}_{\mathcal{M}_i}) + \mathcal{L}_d(\widehat{\mathcal{P}}_i^{\mathcal{X}}, \widehat{\mathbb{P}}_{\mathcal{M}_i}^{\mathcal{X}}) \\
&+ 4q \left( \mathrm{Re}_{U_{\mathcal{P}_i^{\mathcal{X}}}}(\mathcal{H}) + \mathrm{Re}_{U_{\mathbb{P}_{\mathcal{M}_i}^{\mathcal{X}}}}(\mathcal{H}) \right) \\
&+ 3M \left( \sqrt{\frac{\log\left(\frac{4}{\delta}\right)}{2m_{\mathcal{P}_i^{\mathcal{X}}}}} + \sqrt{\frac{\log\left(\frac{4}{\delta}\right)}{2m_{\mathbb{P}_{\mathcal{M}_i}^{\mathcal{X}}}}} \right).,
\end{aligned} \tag{6}$$

This proves Theorem 1.

# B The proof of Theorem 3

**Theorem 3.** For a given data stream $\mathcal{S}$, we assume that $\mathcal{G} = \{G_1, \cdots, G_c\}$ has learned $n$ components at $\mathcal{T}_i$. Let $\mathcal{K} = \{\mathcal{T}_{k_1}, \cdots, \mathcal{T}_{k_c}\}$ be a set of training steps where each $\mathcal{T}_{k_v}$ represents the training step that $G_v$ was finished on training. We derive a GB for $\mathcal{G}$ as :

$$\frac{1}{i} \sum\nolimits_{j=1}^{i} \{\mathcal{R}(h, \mathcal{P}_{(i,j)})\} \leq \frac{1}{i} \sum\nolimits_{j=1}^{i} \{\mathrm{F}_S(\mathcal{P}_{(i,j)}, \mathcal{G})\} \tag{7}$$

where $F_S(\cdot, \cdot)$ is the selection function, defined as :

$$F_S(\mathcal{P}_i, \mathcal{G}) = \min_{v=1,\cdots,c} \left\{ \mathcal{R}(h, h_{\mathcal{M}_{k_i}}, \mathbb{P}_{\mathcal{M}_{k_v}}) \right.$$
$$\left. + \mathcal{L}_{\hat{d}}(\mathcal{P}_i^{\mathcal{X}}, \mathbb{P}_{\mathcal{M}_{k_v}}^{\mathcal{X}}) + \eta(\mathcal{P}_i, \mathbb{P}_{\mathcal{M}_{k_v}}) \right\}, \tag{8}$$

where $\mathbb{P}_{\mathcal{M}_{k_v}}$ represents the memory distribution at $\mathcal{T}_{k_v}$. However, we usually perform the model selection in a batch of samples instead of the whole seen training samples. In the following, we consider $\mathcal{P}_i$ to be a mixture distribution and its density form is expressed by $p_{(1:i)}(\mathbf{x}) = \frac{1}{i}\sum_{j=1}^{i} p_i(\mathbf{x})$, where each $p_i(\mathbf{x})$ is the density function for a batch of samples. $\mathcal{R}(h, \mathcal{P}_i)$ can be expressed as the integral form $\int p_i(\mathbf{x})\mathcal{L}(h, f_{\mathcal{P}_i})\, d\mathbf{x}$. When we take the mixture density form into this integral, we have :

$$\frac{1}{i}\sum_{j=1}^{i} \int p_j(\mathbf{x})\mathcal{L}(h, f_{\mathcal{P}_i})\, d\mathbf{x} \tag{9}$$

Let $\mathcal{P}_{(i,j)}$ be the distribution of $p_j(\mathbf{x})$. Since $f_{\mathcal{P}_i}$ is the true labelling function for $\mathcal{P}_i$, we decompose $f_{\mathcal{P}_i}$ into $j$ functions $\{f_{\mathcal{P}_{(i,1)}}, \cdots, f_{\mathcal{P}_{(i,i)}}\}$ where each $f_{\mathcal{P}_{(i,1)}}$ is the true labelling function for $\mathcal{P}_{(i,j)}$, Then we can rewrite Eq. (9) as :

$$\frac{1}{i}\sum_{j=1}^{i} \int p_j(\mathbf{x})\mathcal{L}(h, f_{\mathcal{P}_{(i,j)}})\, d\mathbf{x} \tag{10}$$

Then we rewrite Eq. (10) as the expectation form $\frac{1}{i}\sum_{j=1}^{i}\{\mathcal{R}(h, \mathcal{P}_{(i,j)})\}$. Then we derive GB for this expectation from as :

$$\frac{1}{i}\sum_{j=1}^{i}\{\mathcal{R}(h, \mathcal{P}_{(i,j)})\} \leq \frac{1}{i}\sum_{j=1}^{i}\{F_S(\mathcal{P}_{(i,j)}, \mathcal{G})\} \tag{11}$$

This proves Theorem 3.

## C  Theoretical analysis for the diversity between components

According to Lemma 2 of the paper, we have the following GB for a dynamic expansion model.

$$\sum_{j=1}^{C_t^T}\left\{\sum_{d=1}^{C_{t,j}^b}\mathcal{R}(h, \mathbb{P}_{t,j}^T(d))\right\} \leq \sum_{j=1}^{C_t^T}\left\{\sum_{d=1}^{C_{t,j}^b}F_S(\mathbb{P}_{t,j}^T(d), \mathcal{G})\right\}. \tag{12}$$

where we assume that $\mathcal{G} = \{G_1, \cdots, G_c\}$ trained $c$ components at $\mathcal{T}_i$. If some of the components in $\mathcal{G}$ learn similar underlying data distributions while $c = C_t^T$, then $\mathcal{G}$ would not capture the information for all target sets $\{\mathcal{D}_{t,1}^T, \cdots, \mathcal{D}_{t,C_t^T}^T\}$, thus deteriorating the performance. On the other hand, as the number of components $c$ in $\mathcal{G}$ increases, $\mathcal{G}$ would learn more knowledge about each target set, which leads to a tight GB in Eq. (12).

A good trade-off between the model's complexity and generalization performance, observed from Eq. (12), is allowing each component to learn the underlying data distribution of a unique target set. Such a mechanism can maintain the model's performance while minimizing the number of components used. To implement this mechanism, we have two main learning goals : (1) Proposing a new approach to detect the data distribution shift in the data stream, providing better signals for the model's expansion. (2) Encouraging the diversity between trained components such that each one models a non-overlapped underlying data distribution. The proposed ODDL-S can satisfy these two learning goals by proposing a new expansion criterion to detect the data distribution shift and a new sample selection to encourage the diversity between components.

In the following, we theoretically prove that the diversity of trained components relieves not only forgetting but also achieves a good generalization performance.

**Assumption 1.** For a given dynamic expansion model $\mathcal{G} = \{\mathcal{G}_1, \cdots, \mathcal{G}_c\}$ that has learnt $c$ components at $\mathcal{T}_i$, let $\mathcal{K} = \{\mathcal{T}_{k_1}, \cdots, \mathcal{T}_{k_c}\}$ be a set of training steps where each $\mathcal{T}_{k_j}$ represents the training step that $G_j$ was trained on. By satisfying the ideal selection process (Eq.(22) of the paper) and also considering that each component $G_t$ finished the training on $\mathcal{M}_{k_t}$ at $\mathcal{T}_{k_t}$, we assume that the dynamic

expansion model $\mathcal{G}$ can be seen as a single model $h$ trained on all previously learnt memories $\{\mathcal{M}_{k_1}, \cdots, \mathcal{M}_{k_{c-1}}\}$ and the current memory $\mathcal{M}_i$ at $\mathcal{T}_i$, where $\mathcal{M}_{k_c} = \mathcal{M}_i$. This assumption is reasonable since the selection process (Eq.(22) of the paper) can always choose a component with the minimal risk. Let $\mathbb{P}_{\bigcup_{t=k_1}^{k_{c-1}} \{\mathcal{M}_t\} \otimes \mathcal{M}_i}$ represent the distribution of all finished memories $\{\mathcal{M}_{k_1}, \cdots, \mathcal{M}_{k_{n-1}}\}$ and the current memory $\mathcal{M}_i$ at $\mathcal{T}_i$.

**Theorem 4.** For a given data stream $\mathcal{S}$, we assume that $\mathcal{G} = \{G_1, \cdots, G_c\}$ has trained $c$ components at $\mathcal{T}_i$. Let $\mathcal{P}_i$ represent the distribution of all visited samples at $\mathcal{T}_i$. Based on Assumption 1. we derive a GB as :

$$
\begin{aligned}
\mathcal{R}(h, \mathcal{P}_i) \leq \mathcal{R}\big(h, h_{\bigcup_{t=k_1}^{k_{c-1}} \{\mathcal{M}_t\} \otimes \mathcal{M}_i}, \mathbb{P}_{\bigcup_{t=k_1}^{k_{c-1}} \{\mathcal{M}_t\} \otimes \mathcal{M}_i}\big) + \mathcal{L}_{\hat{d}}\big(\mathcal{P}_i^{\mathcal{X}}, \mathbb{P}_{\bigcup_{t=k_1}^{k_{c-1}} \{\mathcal{M}_t\} \otimes \mathcal{M}_i}^{\mathcal{X}}\big) \\
+ \eta\big(\mathcal{P}_i, \mathbb{P}_{\bigcup_{t=k_1}^{k_{c-1}} \{\mathcal{M}_t\} \otimes \mathcal{M}_i}\big),
\end{aligned}
\tag{13}
$$

**Remark.** We have several observations from Theorem 4 :

- The forgetting of $h$ on $\mathcal{P}_i$ relies on the discrepancy distance between $\mathcal{P}_i$ and $\mathbb{P}_{\bigcup_{t=k_1}^{k_{c-1}} \{\mathcal{M}_t\} \otimes \mathcal{M}_i}$. Since $\mathbb{P}_{\bigcup_{t=k_1}^{k_{c-1}} \{\mathcal{M}_t\} \otimes \mathcal{M}_i}$ represents the information from all learnt memories and the current memory, the diversity between these memorizes play an important role. For instance, if several memories would capture overlapping underlying data distributions, then this would require to create and add more components to the mixture in order to capture all modes of the statistical data representation $\mathcal{P}_i$. On the other hand, if each memory captures a non-overlapped underlying data distribution, then $\mathbb{P}_{\bigcup_{t=k_1}^{k_{c-1}} \{\mathcal{M}_t\} \otimes \mathcal{M}_i}$ would capture more modes of $\mathcal{P}_i$ using a minimal number of components $c$.
- Compared with Theorem 3 from the paper, Theorem 4 theoretically proves that the probabilistic diversity between trained components in a mixture model is crucial for relieve=ing forgetting.

In the following, we derive a GB to theoretically prove that the diversity between trained components in $\mathcal{G}$ is also important for the generalization performance.

**Theorem 5** For a given data stream $S = \bigcup_{C_t^S}^{j} \mathcal{D}_{t,j}^S$ consisting of samples from $\mathcal{D}_t^S$, we have a set of target sets $\{\mathcal{D}_{t,1}^T, \cdots, \mathcal{D}_{t,C_t^T}^T\}$. Let $\mathbb{P}_t^T$ represent the distribution of all target sets $\{\mathcal{D}_{t,1}^T, \cdots, \mathcal{D}_{t,C_t^T}^T\}$. Based on Assumption 1, we derive a GB as :

$$
\begin{aligned}
\mathcal{R}(h, \mathbb{P}_t^T) \leq \mathcal{R}\big(h, h_{\bigcup_{t=k_1}^{k_{c-1}} \{\mathcal{M}_t\} \otimes \mathcal{M}_i}, \mathbb{P}_{\bigcup_{t=k_1}^{k_{c-1}} \{\mathcal{M}_t\} \otimes \mathcal{M}_i}\big) + \mathcal{L}_{\hat{d}}\big(\mathbb{P}_t^{T,\mathcal{X}}, \mathbb{P}_{\bigcup_{t=k_1}^{k_{c-1}} \{\mathcal{M}_t\} \otimes \mathcal{M}_i}^{\mathcal{X}}\big) \\
+ \eta\big(\mathbb{P}_t^{T,\mathcal{X}}, \mathbb{P}_{\bigcup_{t=k_1}^{k_{c-1}} \{\mathcal{M}_t\} \otimes \mathcal{M}_i}\big),
\end{aligned}
\tag{14}
$$

From Eq. (14), we can observe that as the number of training steps $\mathcal{T}_i$ increases, $\mathcal{G}$ would gain more knowledge and thus gradually improve its generalization performance, as empirically shown in Fig. 2-b from the paper. Since the target distribution $\mathbb{P}_t^T$ involves several different underlying data distributions, the diversity between components in $\mathcal{G}$ helps $\mathbb{P}_{\bigcup_{t=k_1}^{k_{c-1}} \{\mathcal{M}_t\} \otimes \mathcal{M}_i}$ to capture these underlying data distributions with a reasonable number of components.

# D   Forgetting analysis of other methods

In this section, we apply the proposed theoretical framework to analyze the forgetting behaviour of the existing CL methods. To our best knowledge, this paper is the first work to propose a forgetting analysis for continual learning. Moreover, this paper is the first work to bridge the gap between the theory and the existing memory-based approaches for TFCL.

## D.1   Memory-based approaches

Memory-based approaches typically employ a small memory buffer to store a few past samples and replay them when training incoming models. Maximal Interfered Retrieval (MIR), [1] is one of

the most popular memory-based approaches, which uses a memory buffer with a sample selection criterion. We derive the GB from Theorem 1 of the paper for analyzing the forgetting behaviour of MIR :

$$\mathcal{R}(h, \mathcal{P}_i) \leq \mathcal{R}(h, h_{\mathcal{M}_i}, \mathbb{P}_{\mathcal{M}_i}) + \mathcal{L}_{\widehat{d}}(\mathcal{P}_i^{\mathcal{X}}, \mathbb{P}_{\mathcal{M}_i}^{\mathcal{X}}) + \eta(\mathcal{P}_i, \mathbb{P}_{\mathcal{M}_i}). \tag{15}$$

Since the memory buffer in MIR has fixed capacity (the maximum number of stored samples), MIR can obtain a tight GB at several initial training steps ($i$ is small and not larger than the maximum memory size). The main idea of the sample selection in MIR is to encourage storing diverse samples. Since $\mathcal{P}_i$ would involve several underlying data distributions as the number of training steps ($i$) increases, the diversity in the memory plays an important role to ensure a tight GB in Eq. (15). The results of Eq. (15) can also explain the importance of the memory diversity in other memory-based approaches, including the Gradient Sample Selection (GSS) [2] and the Continual Prototype Evolution (CoPE) [4].

### D.2 Generative replay mechanism

We extend the proposed theoretical framework to analyze the forgetting behaviour for the generative replay mechanism based models. Firstly, we introduce several notations as follows.

**Definition 8. (Generative replay mechanism (GRM).)** Let $G$ be single model which consists of a classifier $h \in \mathcal{H}$ and a VAE model $v$. Let $h^i$ and $v^i$ represent the classifier and VAE model updated at the training step $\mathcal{T}_i$. Let $\mathbb{P}_{v^i}$ represent the distribution of generative samples $\{\mathbf{x}, h^i(\mathbf{x})\}, \mathbf{x} \sim v^i$ drawn from $v^i$ and $h^i$. In the following, we derive the GB for GRM-based models based on the results of Theorem 1 :

$$\mathcal{R}(h, \mathcal{P}_i) \leq \mathcal{R}(h, h_{v_i}, \mathbb{P}_{v_i}) + \mathcal{L}_{\widehat{d}}(\mathcal{P}_i^{\mathcal{X}}, \mathbb{P}_{v_i}^{\mathcal{X}}) + \eta(\mathcal{P}_i, \mathbb{P}_{v_i}). \tag{16}$$

where $h_{v_i} = \arg\min_{h \in \mathcal{H}}(h, \mathbb{P}_{v_i})$ represents an ideal classifier for $\mathbb{P}_{v^i}$. From this GB (Eq. (16)), we observe that the quality of generative samples plays an important role for reducing the forgetting. If the generator $v$ can produce more realistic samples such that the discrepancy distance term $\mathcal{L}_{\widehat{d}}(\mathcal{P}_i^{\mathcal{X}}, \mathbb{P}_{v_i}^{\mathcal{X}})$ is small, then the classifier $h$ tends to have a small gap between the target and source risk, thus maintaining the performance on all previously visited samples. In practice, GRM would perform a number of generating processes in which the quality of the data samples generated by the generator retrained on its generations would deteriorate during the following training steps [15, 14], with a higher probabilistic deterioration when the generator is used more often for GRM.

GRM-based approaches for TFCL usually combine the GRM and the memory for the sample replay, such as Maximal Interfered Retrieval (MIR), [1].

**Definition 9. (The joint distribution of the GRM and the memory.)** Let $\mathcal{M}$ be a memory buffer updated at the training step $\mathcal{T}_i$. We can form a joint distribution $\mathbb{P}_{v^i \otimes \mathcal{M}_i}$ from the sampling process $\{\mathbf{x}, h^i(\mathbf{x}), \mathbf{x}', y'\}, \mathbf{x} \sim v^i, \mathbf{x}' \sim \mathcal{M}_i$ at $\mathcal{T}_i$.

From Definition 9, we derive the GB for MIR at $\mathcal{T}_i$ :

$$\mathcal{R}(h, \mathcal{P}_i) \leq \mathcal{R}(h, h_{v_i \otimes \mathcal{M}_i}, \mathbb{P}_{v_i \otimes \mathcal{M}_i}) + \mathcal{L}_{\widehat{d}}(\mathcal{P}_i^{\mathcal{X}}, \mathbb{P}_{v_i \otimes \mathcal{M}_i}^{\mathcal{X}}) + \eta(\mathcal{P}_i, \mathbb{P}_{v_i \otimes \mathcal{M}_i}), \tag{17}$$

where $h_{v_i \otimes \mathcal{M}_i} = \arg\min_{h \in \mathcal{H}}(h, \mathbb{P}_{v_i \otimes \mathcal{M}_i})$ is an ideal classifier for $\mathbb{P}_{v^i \otimes \mathcal{M}_i}$. From Eq. (17), we observe that the forgetting process of MIR relies on the discrepancy distance between $\mathcal{P}_i$ and $\mathbb{P}_{v_i \otimes \mathcal{M}_i}$ at $\mathcal{T}_i$. Although, the GRM adapted in MIR would provide infinite replay samples that relieve the memory limit problem, the quality of generative examples drawn from the GRM is also important for the resulting performance.

### D.3 Dynamic expansion models

Dynamic expansion models (DEMs) have achieved remarkable performances in TFCL. The main advanced properties of DEMs over a single model can be summarized into two aspects. Firstly, DEMs do not suffer from the negative knowledge transfer when choosing a reasonable expansion criterion. However, a single model is prone to degrading the performance if the memory would lose previously visited samples. Secondly, DEMs is scalable for learning an infinite number of data streams.

The existing DEMs [13, 10] evaluate the variance of the loss values as the expansion signals, which has no theoretical guarantees. These approaches also do not analyse the trade-off between the model's

complexity and generalization. Firstly, we derive the GB from Theorem 3 from the paper to analyse the forgetting behaviour of existing DEMs :

$$\frac{1}{i}\sum_{j=1}^{i}\left\{\mathcal{R}\big(h,\mathcal{P}_{(i,j)}\big)\right\} \leq \frac{1}{i}\sum_{j=1}^{i}\left\{\mathrm{F}_S\big(\mathcal{P}_{(i,j)},\mathcal{G}\big)\right\}, \tag{18}$$

$$\mathrm{F}_S\big(\mathcal{P}_{(i,j)},\mathcal{G}\big) = \min_{v=1,\cdots,c}\Big\{\mathcal{R}\big(h,h_{\mathcal{M}_{k_v}},\mathbb{P}_{\mathcal{M}_{k_v}}\big) \tag{19}$$
$$+ \mathcal{L}_{\widehat{d}}\big(\mathcal{P}^{\mathcal{X}}_{(i,j)},\mathbb{P}^{\mathcal{X}}_{\mathcal{M}_{k_v}}\big) + \eta\big(\mathcal{P}_{(i,j)},\mathbb{P}_{\mathcal{M}_{k_v}}\big)\Big\},$$

We only consider an ideal model selection case in Eq. (19). From Eq. (18), it can be observed that by increasing the model's complexity (by increasing the number of components) would lead to a tight GB because more components can capture more information about $\mathcal{P}_i$ during the training. Therefore, the model selection (Eq. (19)) can lead to an appropriate number of components providing minimal risks. In the following, we also investigate the generalization performance of the existing DEMs under TFCL. We derive the GB from Lemma 2 of the paper as :

$$\sum_{j=1}^{C_t^T}\left\{\sum_{d=1}^{C_{t,j}^b}\mathcal{R}\big(h,\mathbb{P}_{t,j}^T(d)\big)\right\} \leq \sum_{j=1}^{C_t^T}\left\{\sum_{d=1}^{C_{t,j}^b}\mathrm{F}_S\big(\mathbb{P}_{t,j}^T(d),\mathcal{G}\big)\right\}. \tag{20}$$

From Eq. (20), it can be observed that additional components would provide better generalization performance on all target distributions because the DEMs capture the underlying data distributions of most target sets $\{\mathcal{D}_{t,1}^T,\cdots,\mathcal{D}_{t,C_t^T}^T\}$. The best configuration for DEMs allows one component to capture the underlying data distribution of a unique target set. In this case, the DEM would achieve a tight GB while minimizing the number of components used. To our best knowledge, this paper is the first research study to provide the theoretical analysis for the forgetting behaviour of DEMs and the trade-off between the model's complexity and its performance.

# E    Additional information for ODDL

The detailed learning procedure of the proposed ODDL-S is presented in Fig. 1 and the algorithm is provided in Algorithm 1.

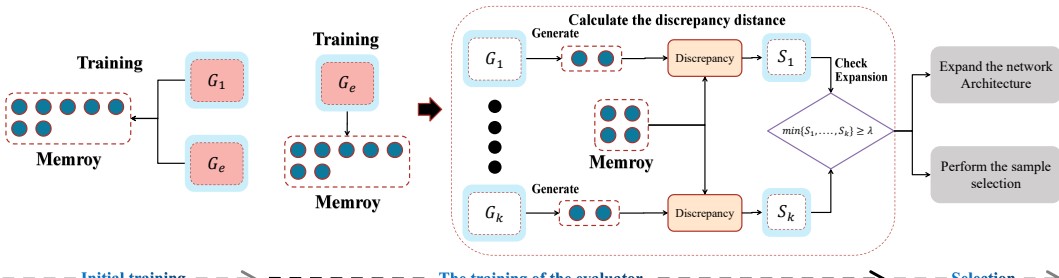

Figure 1: The learning process of the proposed ODDL-S, which consists of three phases. At the initial training phase, we train both $G_1$ and $G_e$. Then we only train $G_e$ when the memory is full while we also check the expansion by using Eq.(18) of the paper. If satisfy the expansion criterion, then we perform the expansion while clearing up the current memory, otherwise, we perform the sample selection (Eq.(20) of the paper).

In addition, we provide supplemental information for the equations (Eq.(16) and Eq.(18)) of the paper. Since Eq.(16) and Eq.(18) of the paper require to access previously learnt memory distribution $\mathbb{P}^{\mathcal{X}}_{\mathcal{M}_{k_1}}$ which is not available during the current training session, we approximate it by using the auxiliary distribution $\mathbb{P}^{\mathcal{X}}_{v_1^j}$ formed by the samples drawn from $v_1^j$ of $G_1^j$. It notes that the number of generative samples matches the number of stored samples in the current memory.

---
**Algorithm 1:** Training of ODDL-S
---
1 **Input**:$\mathcal{S}$, $n$ (Number of training steps);
  1: **for** $i < n$ **do**
  2:    $\{\mathbf{X}_i^b, \mathbf{Y}_i^b\} \sim \mathcal{S}$ ;
  3:    **Initial training stage;**
  4:    **if** (k == 2) **then**
  5:      **if** ($|\mathcal{M}_i| < \mathcal{M}^{Max}$) **then**
  6:        Train the classifiers of $G_1$ and $G_2$ on $\mathcal{M}_i$ using $\mathcal{L}_{class}(G_1, \mathcal{M}_i)$ and $\mathcal{L}_{class}(G_2, \mathcal{M}_i)$;
  7:        Train the VAEs of $G_1$ and $G_2$ on $\mathcal{M}_i$ using $\mathcal{L}_{VAE}(G_1, \mathcal{M}_i)$ and $\mathcal{L}_{VAE}(G_2, \mathcal{M}_i)$;
  8:      **end if**
  9:    **else**
 10:      **Evaluator training stage;**
 11:      **if** ($|\mathcal{M}_i| < \mathcal{M}^{Max}$) **then**
 12:        Train the classifier of $G_e$ on $\mathcal{M}_i$ using $\mathcal{L}_{class}(G_e, \mathcal{M}_i)$;
 13:        Train the VAE of $G_e$ on $\mathcal{M}_i$ using $\mathcal{L}_{VAE}(G_e, \mathcal{M}_i)$;
 14:      **else**
 15:        Calculate the discrepancy distance using Eq.(21) of the paper;
 16:        **if** (satisfy Eq.(22) of the paper **then**
 17:          $\mathcal{G} = G_e \cup \mathcal{G}$ Add the auxiliary component;
 18:          $G_e = G_{k+1}$ Build a new component;
 19:          $k = k + 1$ (Number of components);
 20:        **else**
 21:          **Sample selection stage;**
 22:          $\{\mathbf{x}_1, \cdots, \mathbf{x}_{\mathcal{M}^{Max}}\} \sim \mathcal{M}_i$;
 23:          $\{\mathcal{L}_d^s(\mathcal{G}, \mathbf{x}_1), \cdots, \mathcal{L}_d^s(\mathcal{G}, \mathbf{x}_{\mathcal{M}^{Max}})\}$ (Eq.(23) of the paper)
 24:          $\mathcal{M}_i = \bigcup_{j=1}^{|\mathcal{M}'_i|-b} \mathcal{M}'_i[j]$;
 25:        **end if**
 26:      **end if**
 27:    **end if**
 28: **end for**
---

# F    Additional information for the experiment

**The release of the code.** We have provided the detailed implementation of the proposed approach. We will organize the source code of the ODDL-S model for the sake of easy understanding and for facilitating the re-implementation and we will release it publicly on https://github.com/ if the paper is accepted.

## F.1   Experiment setting

**The hyperparameter configuration and GPU hardware.** To perform the density estimation task, we use Adam [7] with a learning rate of 0.0001 and its default hyperparameters. To perform the generative modelling task, we use the Adam with a learning rate of 0.00005. We set the batch size and the number of epochs for each training step as 64 and 100, respectively. The GPU used for the experiments was GeForce GTX 1080. The operating system considered for experiments was Ubuntu 18.04.5.

**The detained information for the dataset. Split MNIST** divides MNIST [9] containing 60k training samples, into five tasks according to pairs of digits in increasing order [4]. **Split CIFAR10** splits CIFAR10 [8] into five tasks where each task consists of samples from two different classes [4]. **Split CIFAR100** divides CIFAR100 into 20 tasks where each task has 2500 samples from 5 different classes [11].

**The configuration of the network architecture for MNIST.** We adapt the network architecture from [3] where two fully connected layers implement the inference and generator models. Each layer has 200 hidden units. The shared modules use the expansion mechanism as a single fully-connected neural network with a layer (200 hidden units). A single layer also implements each individual

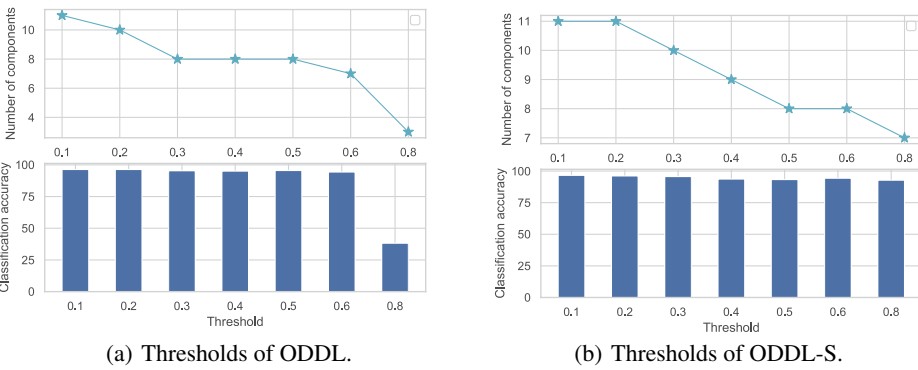

(a) Thresholds of ODDL.        (b) Thresholds of ODDL-S.

Figure 2: The performance change of the proposed approach when varying the threshold.

component with 200 hidden units for both the generator and inference models. We implement the classifier by using an MLP network with 2 hidden layers of 400 units [4] for Split MNIST.

**The configuration of the network architecture for CIFAR10 and CIFAR100.** The shared encoder is implemented using a fully connected network with three layers of processing units [2000, 1500, 1000], and the component encoder uses a fully connected network with three layers [600, 300, 200]. The shared decoder is implemented by a fully connected network with three layers [200, 300, 600] and the component encoder is implemented by a fully connected network with three layers [1000, 1500, 2000]. The dimension of the latent variable is 200. We adapt ResNet 18 [5] as the classifier for Split CIFAR10, Split CIFAR100, and Split MINI-ImageNet.

**Hyperparameters for Split MNIST, Split CIFAR10 and Split CIFAR100 :** Following the setting from [4], we adapt ResNet-18 [5] as the classifier for Split CIFAR10 and Split CIFAR100. We use an MLP network with 2 hidden layers of 400 units [4] as the classifier for Split MNIST. The maximum memory size for Split MNIST, Split CIFAR10, and Split CIFAR100 is 2000, 1000 and 5000, respectively. During the training, we only access a small batch of samples in certain training steps where the batch size is 10. We search $\lambda$ from 0.1 to 4.0 (See Section 4.1 of the paper) and the chosen $\lambda$ values are 0.2, 0.45 and 3.0 for Split MNIST, Split CIFAR10 and Split CIFAR100.

**Additional information for the evaluation.** All results reported in the paper are evaluated on the testing datasets after all training steps finish.

**The details of the large-scale dataset**. Split MINI-ImageNet (Split MImageNet) contains 20 disjoint tasks where each task contains the samples of five classes [1]. Permuted MNIST consists of 10 tasks, where each task assigns a random permutation that is used in the pixel space of all images. After Split MINI-ImageNet learning, the proposed ODDL-S and ODDL built a total six components while CNDPM created ten components.

## F.2   Ablation study

### F.2.1   The effect of the threshold $\lambda$

We investigate the performance of the proposed approach when varying $\lambda$, and the results are reported in Fig. 2. It observes that the small threshold $\lambda$ leads to creating more components for ODDL and ODDL-S while also improving the performance a little. In contrast, a large $\lambda$ discourage the expansion of components. When $\lambda = 0.8$, ODDL only has three components and suffers enormous degenerated performance while ODDL-S does not lose much performance.

### F.2.2   The effect of the memory size

In the following, we investigate the performance change of various models when varying the memory size. We choose the memory size of 100, 200 and 500 for Split MNIST and Split CIFAR10, and the results are presented in Fig. 3a and Fig. 3b. It observes that although the memory size is reduced, the proposed ODDL still outperforms other baselines by a large margin. For Split CIFAR100 and

Split MINI-ImageNet, we choose the memory size of 2000, 5000, and 10000, and the memory size of 5000, 10000, and 20000, respectively. We report the results in Fig. 3c and Fig. 3d, which show that the proposed ODDL achieves the best results in each case and improves the performance by increasing the memory capacity.

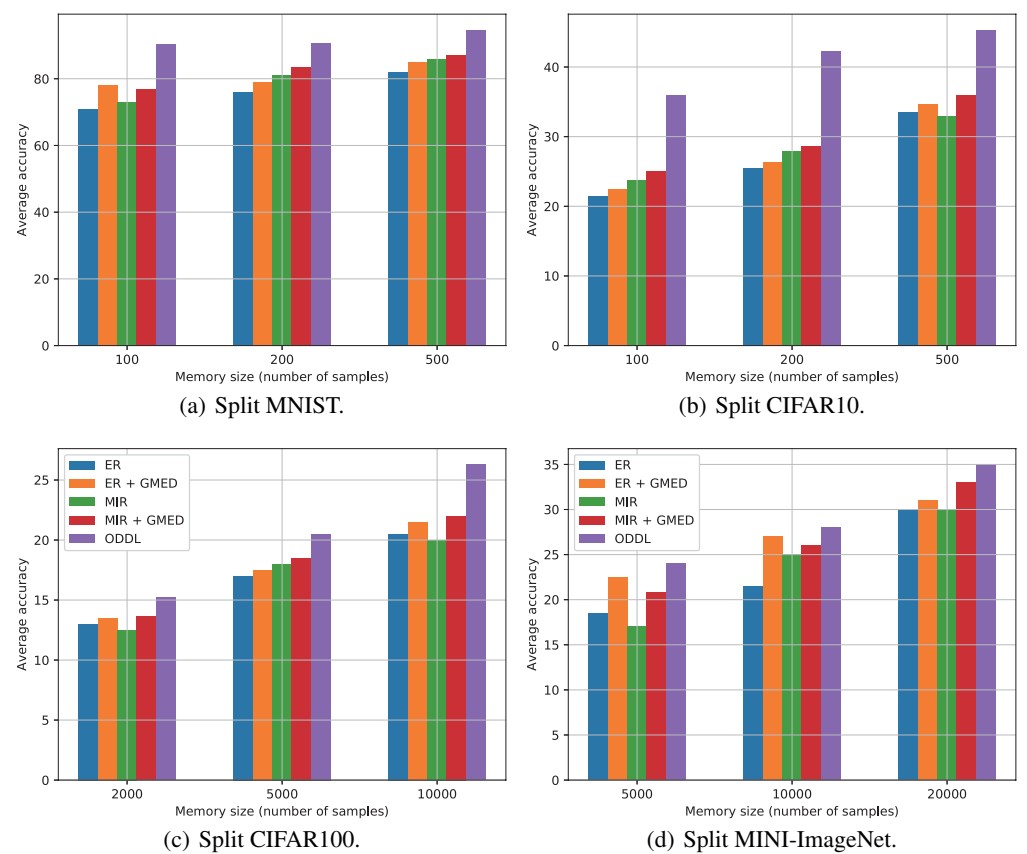

Figure 3: The performance change of various models when changing the memory size.

### F.2.3 The effect of the sample selection in ODDL

We have compared ODDL with the same selection and without. The results are reported in Tab. 1, 2, and 3 of the paper, demonstrating that the proposed sample selection can further improve the performance of ODDL. In addition, we also observe that the sample selection used in ODDL can preserve the performance even if using a very tiny memory buffer, as empirically demonstrated in Fig. 2b. We also compare ODDL with the random-replay memory buffer, namely ODDL-Random. We report the results in Tab. 1, which show that the proposed sample selection used in ODDL outperforms ODDL-Random in each task, demonstrating the effectiveness of the proposed sample selection.

### F.2.4 The effects of the batch size

In this section, we investigate the performance of the proposed ODDL-S when changing the batch size. We train the proposed ODDL-S with different batch sizes on Split MNIST and the results are shown in Fig. 4. It observes that the proposed ODDL-S does not change its performance too much when changing the batch size.

### F.2.5 Detecting the data distribution shift

In the following, we investigate whether the proposed discrepancy-based criterion can provide better signals for the expansion of ODDL-S. We train ODDL-S under Split MNIST and Split CIFAR10

Table 1: The classification accuracy of five indepdnent runs for various models on three datasets. * and † denote the results cited from [4] and [6], respectively.

| Methods | Split MNIST | Split CIFAR10 | Split CIFAR100 |
|---|---|---|---|
| finetune* | $19.75 \pm 0.05$ | $18.55 \pm 0.34$ | $3.53 \pm 0.04$ |
| GEM* | $93.25 \pm 0.36$ | $24.13 \pm 2.46$ | $11.12 \pm 2.48$ |
| iCARL* | $83.95 \pm 0.21$ | $37.32 \pm 2.66$ | $10.80 \pm 0.37$ |
| reservoir* | $92.16 \pm 0.75$ | $42.48 \pm 3.04$ | $19.57 \pm 1.79$ |
| MIR* | $93.20 \pm 0.36$ | $42.80 \pm 2.22$ | $20.00 \pm 0.57$ |
| GSS* | $92.47 \pm 0.92$ | $38.45 \pm 1.41$ | $13.10 \pm 0.94$ |
| CoPE-CE* | $91.77 \pm 0.87$ | $39.73 \pm 2.26$ | $18.33 \pm 1.52$ |
| CoPE* | $93.94 \pm 0.20$ | $48.92 \pm 1.32$ | $21.62 \pm 0.69$ |
| ER + GMED | $82.67 \pm 1.90$ | $34.84 \pm 2.20$ | $20.93 \pm 1.60$ |
| $ER_a$ + GMED | $82.21 \pm 2.90$ | $47.47 \pm 3.20$ | $19.60 \pm 1.50$ |
| CURL* | $92.59 \pm 0.66$ | - | - |
| CNDPM* | $93.23 \pm 0.09$ | $45.21 \pm 0.18$ | $20.10 \pm 0.12$ |
| ODDL | $94.85 \pm 0.02$ | $51.48 \pm 0.12$ | $26.20 \pm 0.72$ |
| ODDL-S | $\mathbf{95.75} \pm 0.05$ | $\mathbf{52.69} \pm 0.11$ | $\mathbf{27.21} \pm 0.87$ |
| ODDL-Random | $95.12 \pm 0.13$ | $51.68 \pm 0.18$ | $20.23 \pm 0.65$ |

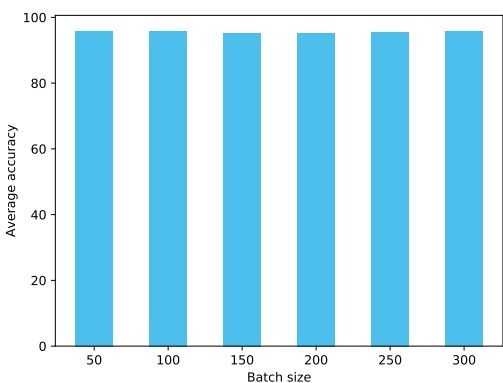

Figure 4: The performance of the proposed ODDL-S when changing the batch size.

where we record the variance of tasks and the number of components in each training step. We plot the results in Fig. 5 where "task" represents the number of tasks in each training step. It observes that the proposed discrepancy-based criterion can detect the data distribution shift accurately, allowing ODDL-S to expand the network architecture when seeing the data distribution shift. This also encourages the proposed ODDL-S to uses the minimal number of components while achieving the optimal performance, as discussed in **Lemma 2** of the paper.

In the following, we also investigate the change of the number of components with respect to the data distribution shift when varying $\lambda$. We plot the results in Fig. 6. It observes that a small $\lambda$ tends to build more components while one or two components model a unique underlying data distribution. Therefore, the best choice of $\lambda$ is 0.5 since it allows ODDL to use the minimal number of components while achieving the optimal performance since each component learns the knowledge from a unique target distribution (task). The reconstruction results of the proposed ODDL-S under Split MNIST is shown in Fig. 7. These results show that the proposed ODDL-S can provide accurate reconstructions for each task/distribution. The generation results of each component after the learning of Split MNIST is shown in Fig. 8 where each component tends to produce different generation results, empirically demonstrating that the proposed ODDL-S can learn a mixture of diverse components during the training.

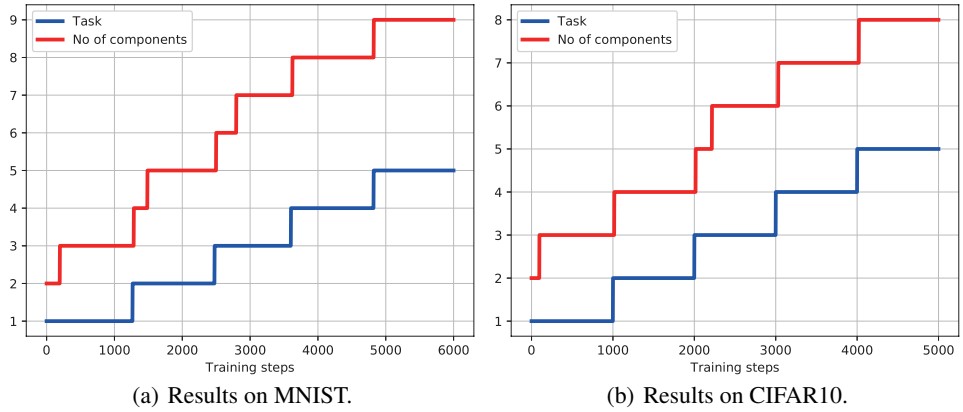

(a) Results on MNIST.                         (b) Results on CIFAR10.

Figure 5: The investigation of the data distribution shift and the change of components in ODDL-S.

### F.2.6    Results on fuzzy task boundaries

We evaluate the effectiveness of the proposed approach on the data stream with fuzzy task boundaries [10] where we exchange the samples between two tasks. We report the results of various models in Tab. 3. These results demonstrate that the proposed ODDL-S outperforms other baselines by a large margin on fuzzy task boundaries.

### F.2.7    Comparison with another sample selection approach in ODDL

We also compare with another sample selection approach in the proposed ODDL, including GSS and reservoir. We apply GSS and reservoir as the sample selection approach in ODDL, namely ODDL-GSS and ODDL-reservoir. We report the classification results in Tab. 2. These results show that the proposed ODDL-S outperforms ODDL-GSS and ODDL-reservoir, which demonstrates the effectiveness of the proposed sample selection approach.

Table 2: The classification accuracy of five independent runs for various models on three datasets. , respectively.

| Methods | Split MNIST | Split CIFAR10 | Split CIFAR100 |
|---|---|---|---|
| ODDL-GSS | $93.46 \pm 0.16$ | $49.26 \pm 0.15$ | $26.79 \pm 0.89$ |
| ODDL-reservoir | $93.16 \pm 0.13$ | $49.26 \pm 0.14$ | $26.25 \pm 0.67$ |
| ODDL | $94.85 \pm 0.02$ | $51.48 \pm 0.12$ | $26.20 \pm 0.72$ |
| ODDL-S | $\mathbf{95.75} \pm 0.05$ | $\mathbf{52.69} \pm 0.11$ | $\mathbf{27.21} \pm 0.87$ |
| ODDL-Random | $95.12 \pm 0.13$ | $51.68 \pm 0.18$ | $20.23 \pm 0.65$ |

Table 3: The classification accuracy of five indepdnent runs for various models over data streams with fuzzy task boundaries.

| Methods | Split MNIST | Split CIFAR10 | Split MImageNet |
|---|---|---|---|
| Vanilla | $21.53 \pm 0.1$ | $20.69 \pm 2.4$ | $3.05 \pm 0.6$ |
| ER | $79.74 \pm 4.0$ | $37.15 \pm 1.6$ | $26.47 \pm 2.3$ |
| MIR | $84.80 \pm 1.9$ | $38.70 \pm 1.7$ | $25.83 \pm 1.5$ |
| ER + GMED | $82.73 \pm 2.6$ | $40.57 \pm 1.7$ | $28.20 \pm 0.6$ |
| MIR+GMED | $86.17 \pm 1.7$ | $41.22 \pm 1.1$ | $26.86 \pm 0.7$ |
| ODDL | $94.25 \pm 0.9$ | $50.07 \pm 1.2$ | $27.98 \pm 1.3$ |
| ODDL-S | $\mathbf{95.55} \pm 1.2$ | $\mathbf{52.27} \pm 2.5$ | $\mathbf{29.76} \pm 1.6$ |

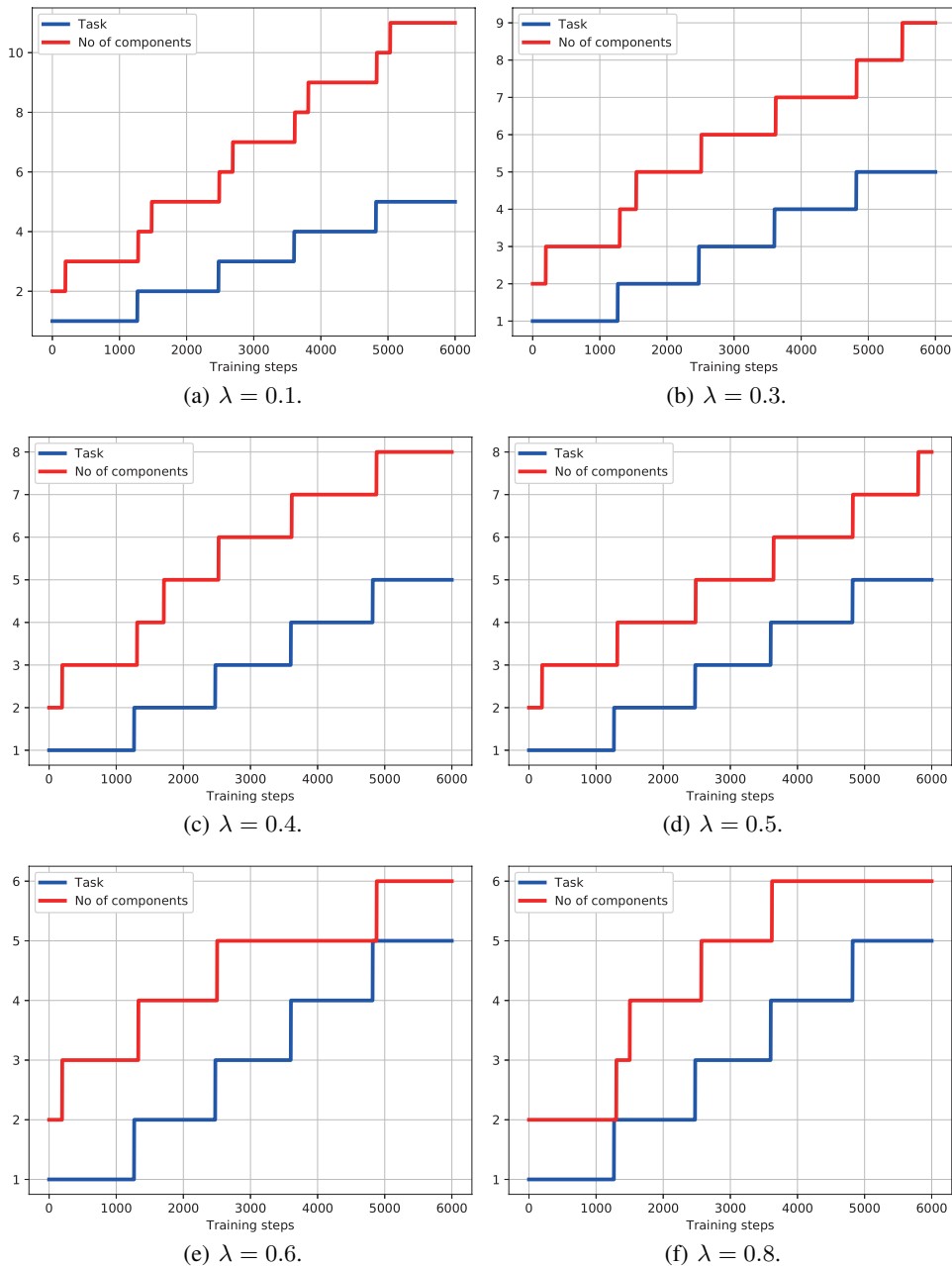

Figure 6: The the change of components in ODDL-S when varying $\lambda$ under Split MNIST.

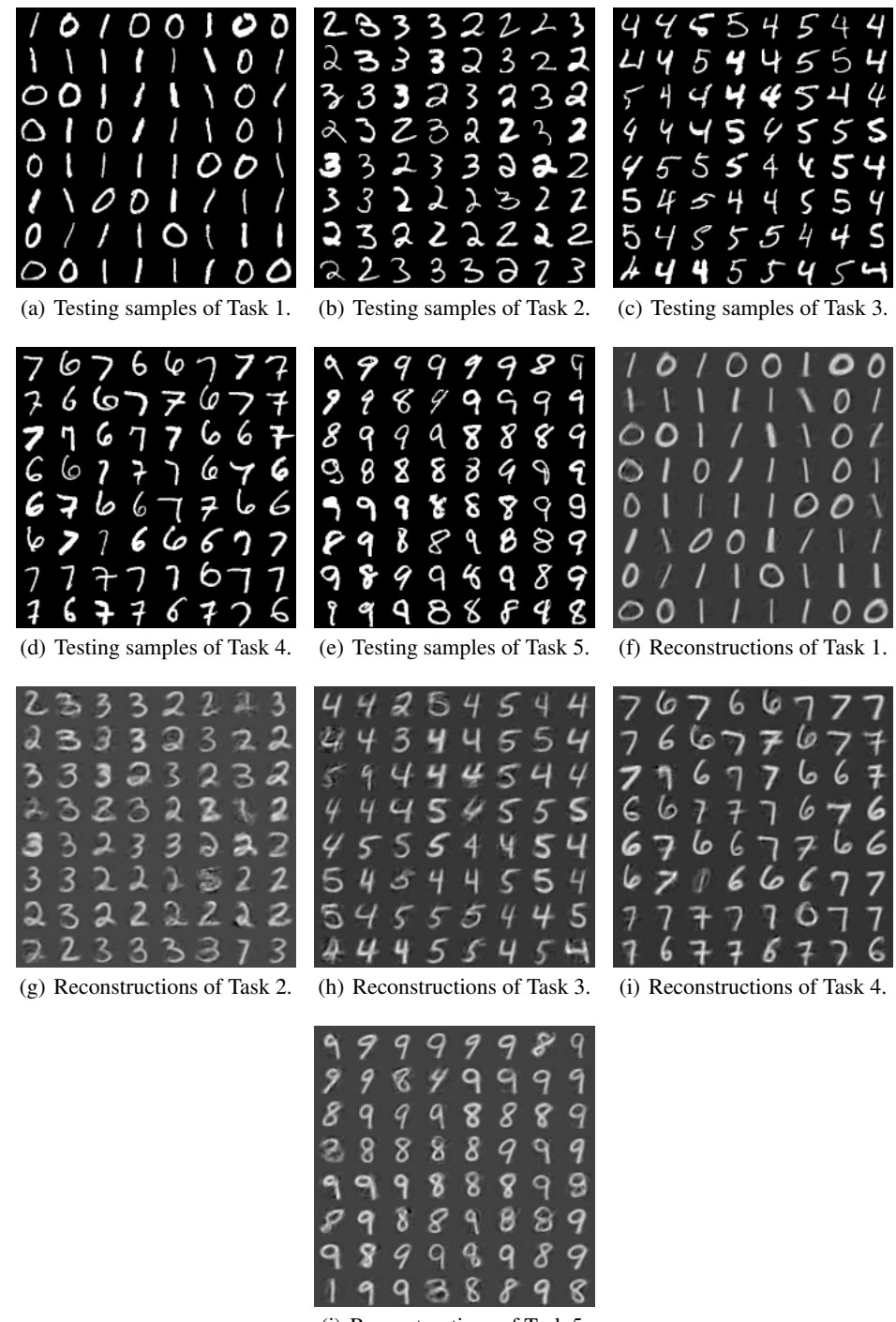

(a) Testing samples of Task 1.    (b) Testing samples of Task 2.    (c) Testing samples of Task 3.

(d) Testing samples of Task 4.    (e) Testing samples of Task 5.    (f) Reconstructions of Task 1.

(g) Reconstructions of Task 2.    (h) Reconstructions of Task 3.    (i) Reconstructions of Task 4.

(j) Reconstructions of Task 5.

Figure 7: The results testing samples and reconstruction results on Split MNIST, achieved by the proposed ODDL-S.

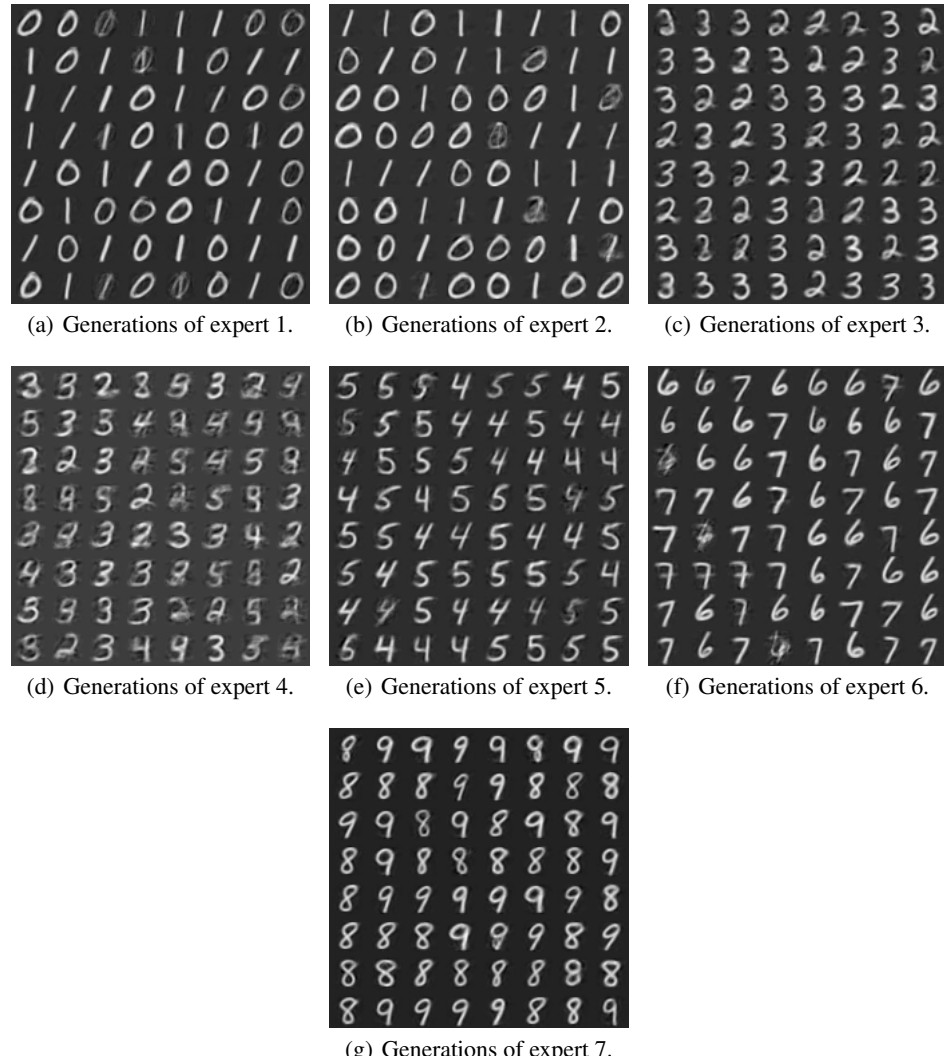

(a) Generations of expert 1.     (b) Generations of expert 2.     (c) Generations of expert 3.

(d) Generations of expert 4.     (e) Generations of expert 5.     (f) Generations of expert 6.

(g) Generations of expert 7.

Figure 8: The generation results by the selected expert of the ODDL-S on Split MNIST.

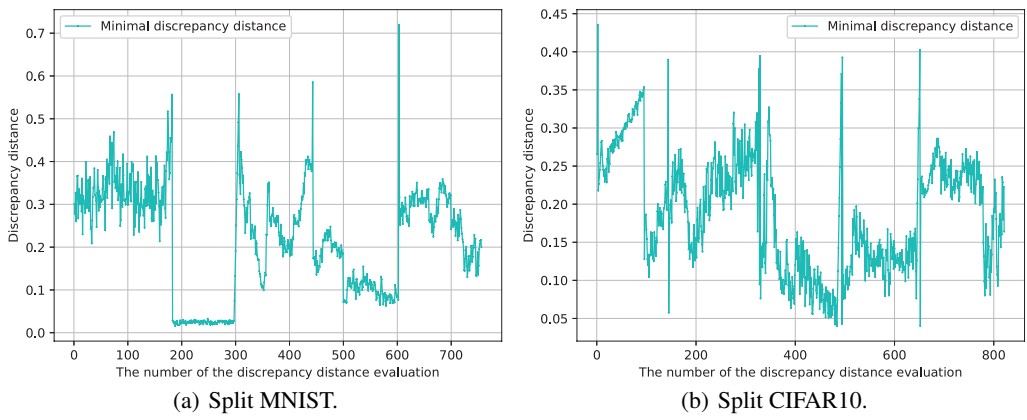

(a) Split MNIST.                  (b) Split CIFAR10.

Figure 9: The estimation of the minimal discrepancy distance under Split MNIST and Split CIFAR10.

### F.2.8 Estimation of the discrepancy distance

We also investigate the minimal discrepancy distance during the training. We train the proposed ODDL-S under Split MNIST and Split CIFAR10, in which we estimate $\min\{\mathcal{L}_d^\star(\mathbb{P}_{v_j^{k_j}}^{\mathcal{X}}, \mathbb{P}_{\mathcal{M}_i}) \mid j = 1, \cdots, s\}$ when checking the expansion criterion (Eq.(20) of the paper). We plot the results in Fig. 9, which show that the minimal discrepancy distance keeps stable in a certain training session and is suddenly becoming large in a certain training step where the incoming samples would draw from a different underlying data distribution. Together with results from Fig. 6, the discrepancy distance has shown to be an effective approach to detect the data distribution shift in the data stream.

### F.2.9 The computational cost of the proposed model

We explain the computational complexity of the model as follows. When the memory buffer is not full, the computational costs of ODDL are only about the training of the current expert because we do not perform the sample selection and dynamic expansion evaluation. In contrast, when the memory buffer is full, the sample selection and dynamic expansion evaluation are performed in each training step, leading to additional computational costs. In addition, increasing the number of components in the ODDL would also slightly increase the computational costs of the sample selection and dynamic expansion evaluation. Therefore, as learning more components over time, the model would increase the computational costs slightly. However, we can accelerate the proposed sample selection and dynamic expansion evaluation by recording the discrepancy score for the memorized samples and only calculating the discrepancy score for each incoming sample once.

We report the training times for ODDL-S and CNDPM in Tab. 4. These results show that the proposed ODDL-S only requires a bit more computational cost compared with CNDPM.

Table 4: The training time (minutes) of various models.

| Methods | Split MNIST | Split CIFAR10 | Split CIFAR100 |
|---------|-------------|---------------|----------------|
| ODDL-S  | 1.2         | 22.2          | 33.68          |
| CNDPM   | 0.9         | 18.6          | 30.23          |

### F.3 Model's parameters

Since only CN-DPM [10] reports the number of parameters for the classification task under TFCL, we provide the comparison on the number of parameters in Table 5. We can observe from this Table that the proposed approach outperforms CN-DPM while requiring fewer parameters.

Table 5: The number of parameters for the classification task. The number of parameters for CN-DPM is reported in [10].

| Methods | Split MNIST | Split CIFAR10 | Split CIFAR100 |
|---------|-------------|---------------|----------------|
| CN-DPM [10] | 524K | 4.60M | 19.2M |
| ODDL-S | 490K | 3.52M | 16.56M |

## G  Negative societal impact and limitation

One potential negative societal impact is that the proposed model could be applied for learning any data streams without checking the data sources, which would lead to the data privacy issue.

One limitation of the proposed model is that the number of parameters would be linearly grown when learning infinite data streams. However, the proposed dynamic expansion criterion and sample selection can allow a newly created component to lean a different underlying data distribution, which reduces the total number of parameters.