# OpenReview forum: "Task-Free Continual Learning via Online Discrepancy Distance Learning"
_NeurIPS.cc/2022/Conference — NeurIPS 2022 Accept_

### Official Review · Reviewer_6NW9 · 2022-07-08

**Rating:** 6
**Confidence:** 3
**Soundness:** 3 good
**Presentation:** 3 good
**Contribution:** 3 good

**Summary:**

The paper proposes a theoretical analysis framework for task free continual learning. The presented framework  gives generalization bounds in terms on discrepancies between distributions. Then, the authors propose an algorithm for task free continual learning that maintains samples in a memory buffer and selects the samples measuring the discrepancy between distributions.

**Questions:**

In numerical experiments, the proposed algorithms is compared with continual learning methods, is it fair the comparison with methods that are not designed for task free continual learning?

You evaluate the method on datasets where the distribution changes in a different manner. For example, in the Split CIFAR10 dataset each task contains new classes, while in the Permuted MNIST dataset each task assigns a random permutation. For which scenarios would it be more appropriate to use your algorithm? For example, for those where the distribution changes smoothly, scenarios where only the joint distribution of labels and instances changes, scenarios where the conditional distribution changes?

It is common in continual learning to assume that tasks are iid or have a common representation, would it make sense to compare your algorithm with methods designed for supervised classification under concept drift? As well as using datasets with concept drift?

Increasing the complexity of the model would lead to tighter bounds, but what is the model's complexity?

Can the bounds presented be compared with those described in papers designed for scenarios with changing distribution? For instance, “New analysis and algorithm for learning with drifting  distributions.”, “Lifelong learning with non-i.i.d. tasks”, or “Minimax Classification under Concept Drift with Multidimensional Adaptation and Performance Guarantees”.

**Limitations:**

Yes

**Strengths And Weaknesses:**

The paper is well written. There are points that should be included in the paper. For example, explain the relationship of the presented bounds with existing bounds for scenarios with changing distribution over time, compare with more methods designed for task free continual learning, and describe the complexity of the method. I include the questions in the following section.

There are some typos in the paper:
	- The authors use the acronym for the supplementary material (SM) in line 147 and define the acronym in line 160.
	- Figure 1 appears far from the text.
	- There is a repeated "the" in line 320

---

> ### Author Response · Authors · 2022-08-02
> **Author Response (Comment 1)**
>
> We sincerely appreciate the reviewer for the thoughtful feedback. Below, we address all comments and questions.
>
> **Comment 1**. There are points that should be included in the paper. For example, explain the relationship of the presented bounds with existing bounds for scenarios with changing distribution over time, compare with more methods designed for task free continual learning, and describe the complexity of the method.
>
> **Answer 1**. Due to the limitation of the space of the current version, we have added the explanation of the relationship of the presented bounds with existing bounds in Appendix-H,2 (Please see the newly uploaded supplemental material). Please also see Answer 7 to Reviewer 6NW9 further down.
>
> To compare more task-free methods, we have adopted the latest TFCL benchmark from [1], published in ICCV 2021 (October), and we believe that we have compared with the most popular TFCL methods published before the submission deadline of NeurIPS 2022. In addition, we also search the google scholar and find that the Dynamic-OCM [7] would be the latest TFCL method, which also uses the same setting from [1]. We compare our model with Dynamic-OCM [7] in the following table. These results show that our model still achieves the best performance on all datasets.
>
>     Table 1. Classification accuracy of five independent runs
>     for various models on three datasets. respectively
> | Methods         | Split MNIST  | Split CIFAR10 | Split CIFAR100 |
> |-----------------|--------------|---------------|----------------|
> | Dynamic-OCM [7] | 94.02 ± 0.23 | 49.16 ± 1.52  | 21.79 ± 0.68   |
> | ODDL-S (our)    | 95.75 ± 0.05 | 52.69 ± 0.11  | 27.21 ± 0.87   |
>
>
> We explain the computational complexity of the model as follows. When the memory buffer is not full, the computational costs of ODDL are only about the training of the current expert because we do not perform the sample selection and dynamic expansion evaluation. In contrast, when the memory buffer is full, the sample selection and dynamic expansion evaluation are performed in each training step, leading to additional computational costs. In addition, increasing the number of components in the ODDL would also slightly increase the computational costs of the sample selection and dynamic expansion evaluation. Therefore, as learning more components over time, the model would increase the computational costs slightly. However, we can accelerate the proposed sample selection and dynamic expansion evaluation by recording the discrepancy score for the memorized samples and only calculating the discrepancy score for each incoming sample once.
>
>
> ## Reference
>
> - [1] Matthias De Lange, and Tinne Tuytelaars. "Continual prototype evolution: Learning online from non-stationary data streams." Proc. of the IEEE/CVF Int. Conf. on Computer Vision (ICCV). 2021.
>
> - [2] Rahaf Aljundi, Klaas Kelchtermans, and Tinne Tuytelaars. "Task-free continual learning." Proc. of the IEEE/CVF Conference on Computer Vision and Pattern Recognition (CVPR). 2019.
>
> - [3] Jin, Xisen, et al. "Gradient-based Editing of Memory Examples for Online Task-free Continual Learning." Advances in Neural Information Processing Systems 34 (2021): 29193-29205.
>
> - [4] Mohri, Mehryar, and Andres Muñoz Medina. "New analysis and algorithm for learning with drifting distributions." Int. Conf. on Algorithmic Learning Theory. Springer, Berlin, Heidelberg, 2012.
>
> - [5]  Anastasia Pentina and Christoph H. Lampert. "Lifelong learning with non-iid tasks." Advances in Neural Information Processing Systems 28 (2015).
>
> - [6] Verónica Álvarez,  Santiago Mazuelas, and Jose A. Lozano. "Minimax Classification under Concept Drift with Multidimensional Adaptation and Performance Guarantees." International Conference on Machine Learning (ICML) (2022).
>
> - [7] Fei Ye and Adrian G. Bors. "Continual Variational Autoencoder Learning via Online Cooperative Memorization." arXiv preprint arXiv:2207.10131, (2022).

---

> > ### Comment · Reviewer_6NW9 · 2022-08-10
> > **Changes**
> >
> > I thank the authors for their response. The response provided some clarifications and the authors include those in the final version of the paper. Therefore, I have decided to change my ranking from a 4 to a 6.
> >
> > The reviewer

---

> ### Author Response · Authors · 2022-08-02
> **Author Response (Comment 2-4)**
>
>
> **Comment 2**. There are some typos in the paper: - The authors use the acronym for the supplementary material (SM) in line 147 and define the acronym in line 160. - Figure 1 appears far from the text. - There is a repeated "the" in line 320.
>
> **Answer 2**. We have revised the entire paper and made the corrections in the revision according to the reviewer's comments. The description of Figure 1 (now is Figure 2 in the revision) is provided in Section 5.1 "Empirical results for the forgetting analysis" of the paper.
>
> To better understand Figure 1 (now is Figure 2 in the revision), we provide more explanations in the following. Figure 1a shows the risk curve of a single model (baseline) and the proposed ODDL on all visited samples and the memory buffer, which provide empirical analysis for Theorem 1 of the paper. The results show that the source risk (risk on the memory buffer) of the single model and ODDL always keep stable during the training because all models can access the samples from the memory buffer. In addition, the single model's target risk (risk on all visited samples) is small for a few initial training steps since the memory can capture all information of visited samples. However, as the number of training steps grows, the target risk of the single model increases, which is caused by the memory that loses previous samples, theoretically explained in Theorem 1. In contrast, the proposed ODDL always keeps a tiny target and source risk during the training, which indicates less forgetting.
>
> Figure 2b (formerly 1b) describes the risk curve of the single model and ODDL on all testing samples and the memory buffer, which provides empirical analysis for Lemma 1 of the paper. These results show that as the number of training steps increases, the single model always keeps a large target risk (risk on testing samples) in the subsequent training steps. In contrast, the proposed ODDL gradually decreases the target risk as the number of training steps increases, as shown in the red line of results from Figure 1b, demonstrating that the proposed ODDL can accumulate knowledge and improve its performance on testing samples over time compared with the baseline.
>
> **Comment 3**. In numerical experiments, the proposed algorithms is compared with continual learning methods, is it fair the comparison with methods that are not designed for task free continual learning?
>
> **Answer 3**. The experiment of this paper adopts the latest TFCL benchmark from the paper [1], published in ICCV 2021, October, and the performance of all baselines in our experiments are cited from [1]. Therefore, we believe that it is a fair comparison.
>
> **Comment 4**. You evaluate the method on datasets where the distribution changes in a different manner. For example, in the Split CIFAR10 dataset each task contains new classes, while in the Permuted MNIST dataset each task assigns a random permutation. For which scenarios would it be more appropriate to use your algorithm? For example, for those where the distribution changes smoothly, scenarios where only the joint distribution of labels and instances changes, scenarios where the conditional distribution changes?
>
> **.Answer 4**. According to the results we consider that our model performs well on both the short and long data streams according to the results from Tables 1 and 2 from the paper. Since the proposed model can dynamically expand its capacity over time, we think the ODDL is more appropriate to deal with a very long data stream involving several data distributions.
>
> 1. For the scenario where the distribution changes smoothly, we can use a suitable $\lambda$ in the dynamic expansion mechanism (Eq.(16) of the paper) to detect the data distribution shift, which allows the ODDL model to appropriately expand the network architecture when needed. Our experimental results from the paper and from the supplementary material indicate the effectiveness of the proposed methodology when learning series of databases under various learning circumstances:
>
> 2. For the scenario when only the joint distribution of labels and instances changes, we can employ a very small $\lambda$ in the proposed dynamic expansion mechanism to appropriately expand the network architecture.
>
> 3. For the scenario where the conditional distribution changes, our model also performs well on the Permuted MNIST. In addition, the proposed ODDL performs well in a more challenging setting, called fuzzy task boundaries, where we exchange the samples between two tasks [1] (Please see results in Appendix-F.2.6 from the supplemental material).
>
> In summary, we believe that our model is more suitable for the standard TFCL scenarios where each distribution contains new classes. Nevertheless, with small changes in the procedure for deciding new components, the proposed methodology can be easily adapted for any other lifelong learning scenario.

---

> ### Author Response · Authors · 2022-08-02
> **Author Response (Comment 5-6)**
>
>
> **Comment 5**. It is common in continual learning to assume that tasks are iid or have a common representation, would it make sense to compare your algorithm with methods designed for supervised classification under concept drift? As well as using datasets with concept drift?
>
> **Answer 5**. In the Supervised Learning Under Concept Drift (SLUCD) [4], instance-label pairs are often changed over time. During the training, the algorithm requires to update its previous classification rule when learning a new sample [4]. To compare to SLUCD, the Task-Free Continual Learning (TFCL) [2] also assumes that the instances would be changed over time. Unlike SLUCD, the model in TFCL can not access all previously seen samples during the training. Therefore, the learning goal in TFCL mainly focuses on addressing the model forgetting while SLUCD is more focused on the adaptation of a new learning environment. Therefore, we believe that TFCL and SLUCD would have some connections but are not the same topic. In addition, the experiment setting and datasets used in TFCL and SLUCD are sufficiently different. We believe that it would not be suitable for comparing our model directly with the existing algorithm designed for SLUCD.
>
> We also believe that learning several datasets with concept drift would be an interesting topic in SLUCD. However, due to the different experiment settings and datasets between TFCL and SLUCD, it would not be suitable for comparing our model with the existing SLUCD algorithm under this setting.
>
> **Comment 6**. Increasing the complexity of the model would lead to tighter bounds, but what is the model's complexity?
>
> **Answer 6**. The model's complexity denotes the number of components in our theoretical analysis. In Lemma 2 of the paper, we show that by increasing the number of components would improve the performance. In addition, we provide the theoretical analysis for the diversity between components in Appendix-C from the supplemental material, demonstrating that maintaining the knowledge diversity among components can lead to an optimal trade-off between the model complexity and generalization performance.
>
> ----
>
> ## Reference
>
> - [1] Matthias De Lange, and Tinne Tuytelaars. "Continual prototype evolution: Learning online from non-stationary data streams." Proc. of the IEEE/CVF Int. Conf. on Computer Vision (ICCV). 2021.
>
> - [2] Rahaf Aljundi, Klaas Kelchtermans, and Tinne Tuytelaars. "Task-free continual learning." Proc. of the IEEE/CVF Conference on Computer Vision and Pattern Recognition (CVPR). 2019.
>
> - [3] Jin, Xisen, et al. "Gradient-based Editing of Memory Examples for Online Task-free Continual Learning." Advances in Neural Information Processing Systems 34 (2021): 29193-29205.
>
> - [4] Mohri, Mehryar, and Andres Muñoz Medina. "New analysis and algorithm for learning with drifting distributions." Int. Conf. on Algorithmic Learning Theory. Springer, Berlin, Heidelberg, 2012.
>
> - [5]  Anastasia Pentina and Christoph H. Lampert. "Lifelong learning with non-iid tasks." Advances in Neural Information Processing Systems 28 (2015).
>
> - [6] Verónica Álvarez,  Santiago Mazuelas, and Jose A. Lozano. "Minimax Classification under Concept Drift with Multidimensional Adaptation and Performance Guarantees." International Conference on Machine Learning (ICML) (2022).
>
> - [7] Fei Ye and Adrian G. Bors. "Continual Variational Autoencoder Learning via Online Cooperative Memorization." arXiv preprint arXiv:2207.10131, (2022).

---

> ### Author Response · Authors · 2022-08-02
> **Author Response (Comment 7)**
>
>
> **Comment 7**. Can the bounds presented be compared with those described in papers designed for scenarios with changing distribution? For instance, “New analysis and algorithm for learning with drifting distributions.”, “Lifelong learning with non-i.i.d. tasks”, or “Minimax Classification under Concept Drift with Multidimensional Adaptation and Performance Guarantees”.
>
> **Answer 7**. In the following we discuss the difference between the proposed bound and bounds provided in [4, 5, 6], which are indicated by the reviewer.
>
> 1. The bound (Theorem 1) in [4] assumes that the model is trained on all previous samples, which is not practical under a TFCL setting. In contrast, in Theorem 1 of our theory, the model is trained on the fixed-length memory buffer, which can be used in the context of TFCL. In addition, the bounds in [4] mainly provide the theoretical guarantees for the performance of a new data distribution (LHS of Theorem 1 of [4]), which does not provide an analysis of the model forgetting. In contrast, our theory is mainly focusing on the forgetting analysis by our model. Moreover, our  theory is easily to be extended for analyzing the forgetting behaviour of a variety of continual learning methods while this cannot be said about the study from [4] (Please see Appendix-D from supplemental material).
>
> 2. When comparing with [5], our theoretical analysis does not rely on the explicit task boundary, which is a more realistic assumption for TFCL. In addition, [5] employs the KL divergence to measure the distance between two distributions, which would require knowing the explicit distribution form and is thus hard to estimate in practice during the learning. However, our theory employs the discrepancy distance, which is easier and can be estimated more reliably. Moreover, the study from [5] does not develop a practical algorithm to be used according to the theoretical analysis. In contrast, our theory provides guidelines for the algorithm design under the TFCL assumption. Finally, inspired by the proposed theoretical analysis, this paper develops a successful continual learning approach for TFCL.
>
> 3. Theorem 1 in [6], similar to [4], assumes that the model is trained on all uncertain data sets over time. However, the number of uncertain data sets would grow infinitely over time, which would not be suitable for TFCL since TFCL usually has restrictions on storing past samples, such as the maximum memory buffer size [1]. In contrast, Theorem 1 in our theory represents a realistic TFCL scenario in which the model is trained on a fixed-length memory buffer and is evaluated on all previously seen samples. Therefore,  the analysis for the forgetting behaviour of the model under TFCL relies on Theorem 1 and its consequences. In addition,  the study from [6], similarly to [4,5],  only provides a theoretical guarantee for a single model. In contrast, we extend our theoretical analysis to the dynamic expansion model, which motivates us to develop a novel continual learning approach for TFCL. Moreover, this paper is the first work to provide the theoretical analysis for the diversity of knowledge recorded by different components (Please see Appendix-C from supplemental material). Such analysis indicates that by maintaining the knowledge diversity among components we ensure a good trade-off between the model's complexity and generalization performance, providing invaluable insights into the algorithm design for TFCL.
>
> Due to the page limitation of the current paper, we have added the discussion between the proposed bound and existing bounds in Appendix-H.2 from the newly uploaded supplemental material.
>
> ----
>
> ## Reference
>
> - [1] Matthias De Lange, and Tinne Tuytelaars. "Continual prototype evolution: Learning online from non-stationary data streams." Proc. of the IEEE/CVF Int. Conf. on Computer Vision (ICCV). 2021.
>
> - [2] Rahaf Aljundi, Klaas Kelchtermans, and Tinne Tuytelaars. "Task-free continual learning." Proc. of the IEEE/CVF Conference on Computer Vision and Pattern Recognition (CVPR). 2019.
>
> - [3] Jin, Xisen, et al. "Gradient-based Editing of Memory Examples for Online Task-free Continual Learning." Advances in Neural Information Processing Systems 34 (2021): 29193-29205.
>
> - [4] Mohri, Mehryar, and Andres Muñoz Medina. "New analysis and algorithm for learning with drifting distributions." Int. Conf. on Algorithmic Learning Theory. Springer, Berlin, Heidelberg, 2012.
>
> - [5]  Anastasia Pentina and Christoph H. Lampert. "Lifelong learning with non-iid tasks." Advances in Neural Information Processing Systems 28 (2015).
>
> - [6] Verónica Álvarez,  Santiago Mazuelas, and Jose A. Lozano. "Minimax Classification under Concept Drift with Multidimensional Adaptation and Performance Guarantees." International Conference on Machine Learning (ICML) (2022).
>
> - [7] Fei Ye and Adrian G. Bors. "Continual Variational Autoencoder Learning via Online Cooperative Memorization." arXiv preprint arXiv:2207.10131, (2022).

---

### Official Review · Reviewer_VJEH · 2022-07-12

**Rating:** 7
**Confidence:** 3
**Soundness:** 4 excellent
**Presentation:** 2 fair
**Contribution:** 3 good

**Summary:**

- The authors derive the first generalisation bound for Task Free Continual Learning. The analysis is inspired by domain adaptation theory.
- The authors leverage the discrepancy measure to derive a continual learning algorithm which balances between model complexity, generalisation and catastrophic forgetting.
- In order to do so, the model is designed as a mixture of smaller classifiers. To each classifier is attached a VAE. The VAE is used to flush the memory and model selection. At test time, the generated samples by the VAE are used to compute the discrepancy with the test samples, in order to identify which model to use for inference.

**Questions:**

- In the appendix, Table 3, it looks like the average accuracy gain with the method decreases with the complexity or scale of the dataset. Do you have any intuition why the improvement on Split MImageNet is much smaller than for CIFAR-10 and Split-MNIST ?
- Do you have any indications for practitioners on how to set the threshold lambda ? Specifically in the case of very long streams, where the optimal lambda may vary over time.

A few typos :
- Replace "CIFRA" by "CIFAR"
- Replace "Memroy" by "Memory" in Figure 2

**Limitations:**

The limitations and potential negative societal impact are discussed adequately in the section G of the Appendix.

**Strengths And Weaknesses:**

## Originality
### 1- Are the tasks or methods new?
The following elements are new :
- The generalisation bound for Task-Free Continual Learning.
- Leveraging the tools of domain adaptation theory for continual learning.
- The final algorithm ODDL with the model expansion, model selection, sample selection and discrepancy as a primitive.

### 2- Is the work a novel combination of well-known techniques?
- The work builds upon the out of distribution theory, and expands it in meaningful ways to derive an algorithm for Task-free Continual Learning.

### 3- Is it clear how this work differs from previous contributions?
- The way the contributions of this work stand out is clear.

### 4- Is related work adequately cited ?
- To my knowledge, the related work is adequately cited.

## Quality
### 1- Is the submission technically sound ?
- The submission is technically sound.

### 2- Are claims well supported (e.g., by theoretical analysis or experimental results) ?
- The theoretical results are supported by proofs.
-

### 3- Are the methods used appropriate ?
- Using the VAE to "summarise" the distribution of a subcategory looks relevant.
- Also, the way discrepancy distance is used in the algorithm is theoretically motivated.

### 4- Is this a complete piece of work or work in progress ?
- It is a complete piece of work.

### 5- Are the authors careful and honest about evaluating both the strengths and weaknesses of their work ?
- The authors benchmarked their method ODDL against several baselines in Table 1.
- The linear growth of the number of models in case of a very large number of tasks, was briefly discussed in the appendix, section G..

## Clarity:
### 1- Is the submission clearly written?
- I read the submission a few times, in order to understand the high level picture of the algorithm, and how its components fit.
- One way to improve clarity is maybe by improving the diagram in Figure 2, by making it more verbose ?

### 2- Is it well organised? (If not, please make constructive suggestions for improving its clarity.)
- I suggest presenting a big picture of the algorithm first with the main building blocks at a high level, then presenting each element in detail. While reading through section 4.3, I realised I didn't read section 4.1 and 4.2 through the right angle, so I had to go back and forth between both sections a few times to get the right picture.

### 3- Does it adequately inform the reader? (Note that a superbly written paper provides enough information for an expert reader to reproduce its results.)
- The hyperparameters, code and required resources are shared by the authors.

## Significance:
### 1- Are the results important?
- The results are important in the following ways :
   - A theoretically grounded approach for Task-Free Continual Learning.
   - An improvement of 7% of average accuracy against the top baseline, on CIFAR-10 and CIFAR-100.
   - A modular algorithm, which is easy to interpret and debug.

### 2- Are others (researchers or practitioners) likely to use the ideas or build on them?
- Given the improvements over the baselines in the experiments (Table 1), it is likely that practitioners will benchmark this method. However, given that it requires a non-negligible amount of engineering and hyperparameter tuning, it is likely that practitioners will first benchmark simpler to implement methods.
- Given that the way the approach is designed is clearly explained and grounded on theory, it make it more likely that researchers will build upon it.

### 3- Does it advance the state of the art in a demonstrable way?
- The results advance the state of the art by 7% of average accuracy over the top baseline for CIFAR-10 and CIFAR-100. (Table 1)

---

> ### Author Response · Authors · 2022-08-02
> **Author Response (Comment 1-4)**
>
>
> We sincerely appreciate the reviewer for the thoughtful feedback.
>
> **Comment 1**. One way to improve clarity is maybe by improving the diagram in Figure 2, by making it more verbose ?
>
> **Answer 1**.  We have added additional descriptions in Figure 1 (which is the former Figure 2) as described in the following. Figure 1 illustrates the proposed expansion criterion and sample selection of the ODDL model when it already has $s$ components. We generate the knowledge by using the VAE $V_j$ of each previously learnt component $G\_j, j =1, \cdots,s-1$, which is used to evaluate the discrepancy distance $S\_j = {\mathcal{L}}^\star_d \big({\mathbb{P} }^{{\mathcal{X}}}\_{v\_j}, {\mathbb{P}}\_{{\mathcal{M}}\_i} \big)$ at ${\mathcal{T}}\_i$ (Eq.(17) of the paper) between $G_j$ and the memory buffer. Then we use these discrepancy scores $\{S\_1,\cdots,S\_{s-1}\}$ to check the model expansion (Now Eq.(18) of the uploaded revision).
>
> Due to the page limitation, we do not include additional descriptions of Figure 1 in the current revised paper. Therefore, we include more descriptions of Figure 1 in Appendix-H.1 from the updated supplemental material which has been uploaded.
>
> **Comment 2**. I suggest presenting a big picture of the algorithm first with the main building blocks at a high level, then presenting each element in detail. While reading through section 4.3, I realized I didn't read section 4.1 and 4.2 through the right angle, so I had to go back and forth between both sections a few times to get the right picture.
>
> **Answer 2**.  We have moved the first paragraph of Section 4.3 to the beginning of Section 4 in the revised paper. We have also added a new paragraph at the beginning of Section 4.
>
> The newly added paragraph contains the following text.
> The training algorithm for the proposed ODDL consists of three stages (We provide Figure 1 in Appendix-E from the newly uploaded supplemental material). In the starting training stage, we aim to learn the initial knowledge of the data stream and preserve it into a component $G_1$ which is frozen, and which can provide the information for the dynamic expansion and sample selection evaluation in the subsequent learning. The evaluator trains the current component following the valuation of the discrepancy distance between each previously learnt component and the memory buffer, providing appropriate signals for the model expansion. In the sample selection stage, we aim to allow the current component to learn novel samples, which would promote the knowledge diversity among components.
>
>
> **Comment 3**. In the appendix, Table 3, it looks like the average accuracy gain with the method decreases with the complexity or scale of the dataset. Do you have any intuition why the improvement on Split MImageNet is much smaller than for CIFAR-10 and Split-MNIST ?
>
> **Answer 3**. Since Split MImageNet contains images of higher complexity than Split MNIST and Split CIFAR10, the improvement on such dataset consisting of complex images is more challenging. In addition, by applying fuzzy task boundaries in Table 3 of the Appendix. can further increase the complexity of Split MImageNet.
>
> **Comment 4**. Do you have any indications for practitioners on how to set the threshold lambda ? Specifically in the case of very long streams, where the optimal lambda may vary over time.
>
> **Answer 4**. Decreasing the expansion threshold $\lambda$ in (Now Eq.(18) is the uploaded revision) can allow the ODDL to create more components, increasing the number of parameters but improving the performance. In contrast, a large $\lambda$ encourages the ODDL to use fewer components, which would deteriorate the performance. Therefore, we can choose a small $\lambda$ for high performance without considering the model's complexity and a large $\lambda$ for learning a compact model while less focusing on the performance. We usually search $\lambda$ from 0.1 to 0.8 for most datasets, while a suitable $\lambda$ configuration allows the ODDL to achieve good performance with a fair number of components. The effect of the choice of $\lambda$ is investigated in Appendix-F.2.1 from supplemental material.
>
> We think that by varying $\lambda$ during the training is an excellent idea for learning long streams. One way to implement this goal is to employ a counter that records the frequency of the model expansion. If the model does not expand the network architecture for a long time, we dynamically decrease $\lambda$. In contrast, if the model performs the expansion frequently, we dynamically increase $\lambda$. Such a mechanism can avoid the rapid growth of the model's complexity while learning a sufficient number of components to ensure good performance. In addition, as the number of training steps grows, the proposed ODDL can accumulate more knowledge into previously learnt components and thus does not perform the expansion frequently, according to (Now Eq.(18) is the uploaded revision).

---

> ### Author Response · Authors · 2022-08-02
> **Author Response (Comment 5)**
>
>
> **Comment 5**. Replace "CIFRA" by "CIFAR", Replace "Memroy" by "Memory" in Figure 2.
>
> **Answer 5**. We have made the correction on the revision according the reviewer's comments. Please see the uploaded revision.

---

> ### Comment · Reviewer_VJEH · 2022-08-10
> **Thanks for the additional clarifications**
>
> Thanks to the authors for the additional responses and clarifications !
>
> I will keep my rating to 7 (accept).

---

### Official Review · Reviewer_aR21 · 2022-07-15

**Rating:** 7
**Confidence:** 3
**Soundness:** 3 good
**Presentation:** 3 good
**Contribution:** 3 good

**Summary:**

The paper proposes a new theoretical framework to study the task-free continual learning problem. The contribution is twofold:
* An extension of the domain adaptation theory allowing for multiple time steps and yielding time-dependent generalization bounds: This analysis explains the forgetting behavior with the discrepancy distance between the memory (source) and targets domains.
* An algorithm inspired by the theoretical results: The discrepancy distance is used to first design a dynamic expansion mechanism. This distance provides a criterion to detect when to add new components to the model.  It is also used to propose a sample selection procedure that insures probabilistic diversity of the information stored in the memory.
This algorithm is then tested on different benchmarks and compared to prior task-free continual learning approaches, showing a substantial improvement.

**Questions:**

* While the theory has been developed for the task-free case, can it be adapted to also account for the case with known task boundaries?
* It seems in the initial definitions that it is assumed that the data is presented to the model ordered by category. Is this the case? If yes: While this is common in the literature, how important is this assumption to the analysis? Would it be possible to extend the analysis to also hold for the case where the data distribution is shifting in a smoother way (model exposed to a slowly changing mixture of categories for example)?
* Can the authors comment on how he model deals with a case where knowledge previously seen is encountered again after initiating a new model? Given that the memory is erased when a new component is initiated, wouldn't this lead to some redundancy in the learnt models? Would it be possible to add a phase to recombine the knowledge encoded in the different components of the model?
* Can the authors comment of the computational cost of their method, and how the model grows over time, as compared to the prior works?
* The field has long focused on catastrophic forgetting. The analysis in this paper is also built around fighting this behavior. When training on a stream of data, however, one could argue that the present and future performance should have a higher weight. What do the authors think about this? Can their analysis and proposed method be adapted to look into forward transfer rather than preventing the model from forgetting?

**Limitations:**

The authors do not discuss the limitations in the paper. As mentioned above, it would be interesting to consider analysing the computational cost of the proposed method.


**Strengths And Weaknesses:**

Strengths:
* The paper is generally clear and well structured.
* The take of the authors on the problem is novel to the best of my knowledge.
* The theoretical analysis proposed in this paper is sound, well motivated and well structured. It is of significance to the community and can shed the light on novel insights for the challenging problem it is focusing on.
* The algorithm derived from the theory makes sense, and the experiments seem to validate its efficiency.

Weaknesses:
* Minor: There is an overload of abbreviation in the paper (e.g. SM for supplementary materials, GB for generalization bounds seems non-necessary), and some inconsistency in the notations (e.g.: The indices in the preliminary (3.1) seem wrong at some places, tasks are sometime indexed by t and sometimes by i. This can make reading the paper quite hard.
* While the training phases of the expansion model are clear, it is quite unclear how the model is used at inference time. It would be great if this can be made clearer. It seems that the VAE is used to decide which trained component to use. Is this the case?
* While the experiments are diverse and show an improvement over prior works, it is unclear how the methods compare from the computational (and less importantly the memory) cost point of view.

---

> ### Author Response · Authors · 2022-08-02
> **Author Response (Comment 1-6)**
>
> We sincerely appreciate the reviewer for the thoughtful feedback. Below, we address all comments and questions.
>
> **Comment 1**. Minor: There is an overload of abbreviation in the paper (e.g. SM for supplementary materials, GB for generalization bounds seems non-necessary), and some inconsistency in the notations (e.g.: The indices in the preliminary (3.1) seem wrong at some places, tasks are sometime indexed by t and sometimes by i. This can make reading the paper quite hard.
>
> **Answer 1**. Thank you for your remark. We use the abbreviation due to the page limitation of the current version. All these abbreviations are specified when used the first time, such as Supplementary Materials by (SM).
>
> We have reduced the number of abbreviations used in Section 3.1 and carefully checked all notations throughout the paper according to the reviewer's comments (Please see the uploaded revised paper).
>
> **Comment 2**. While the training phases of the expansion model are clear, it is quite unclear how the model is used at inference time. It would be great if this can be made clearer. It seems that the VAE is used to decide which trained component to use. Is this the case?
>
> **Answer 2**. Yes, since VAE can estimate the sample log-likelihood for a given input, we compare the sample log-likelihood estimated by the VAE associated with each component. This criterion guides the model to choose an appropriate component with the highest sample log-likelihood.
>
> **Comment 3**. While the experiments are diverse and show an improvement over prior works, it is unclear how the methods compare from the computational (and less importantly the memory) cost point of view.
>
> **Answer 3** We compare the proposed ODDL and CNDPM on three datasets in terms of the training time. The training time for ODDL on Split MNIST, Split CIFAR10 and Split CIFAR100 is 1.2, 22.2 and 33.68,  respectively, minutes. The training time for CNDPM on Split MNIST, Split CIFAR10 and Split CIFAR100 is 0.9, 18.6 and 30.23, respectively, minutes. This result shows that the proposed ODDL only requires a bit more computational cost compared with CNDPM.
>
> **Comment 4**. While the theory has been developed for the task-free case, can it be adapted to also account for the case with known task boundaries?
>
> **Answer 4**. Theorem 1 of the paper can be extended for analyzing a general continual learning framework where the task information is known. In addition, we can also define the task boundary and the probabilistic representation of each task, which can be used in Lemma 1 and Theorem 3 of the paper.
>
> **Comment 5**. It seems in the initial definitions that it is assumed that the data is presented to the model ordered by category. Is this the case? If yes: While this is common in the literature, how important is this assumption to the analysis?
>
> **Answer 5** Yes, we assume a class-incremental setting in the proposed theory. The importance of this assumption in our analysis can be summarized into three points.
>
> 1. Since the proposed theory mainly focuses on the class-incremental setting which is common and popular in continual learning area, our theoretical analysis can be easily extended for analyzing the forgetting behaviour of a variety of CL methods (See details in Appendix-D from supplemental material).
>
> 2. This assumption can allow us to define the probability representation for each data category, which is mainly used for analyzing the generalization performance of the model under TFCL. For instance, Lemma 1 (Eq.(5)) of the paper indicates that the performance of a single model on all target sets is mainly depending on the discrepancy distance between each target domain (distribution) and the memory distribution. Such an analysis can clearly explain how updating the memory buffer in each training step can affect the model's performance.
>
> 3. This assumption in the proposed theory can provide new insights into how data distribution (category) shift can affect the model's performance in each training step. Based on Lemma 1 and Lemma 2 of the paper, we can empirically investigate the forgetting behaviour of the model in Fig.1b of the paper, which clearly describes the forgetting process of the model.
>
> On the other hand, Theorem 1 of the paper does not need this assumption, which can be used in various continual learning settings.
>
> **Comment 6**. Would it be possible to extend the analysis to also hold for the case where the data distribution is shifting in a smoother way (model exposed to a slowly changing mixture of categories for example)?
>
> **Answer 6**. In Theorem 1 of the paper, we do not require any forms of the data stream ${\mathcal{S}}$ and define $\mathcal{P}_i$ to represent the distribution of all visited training samples (including all previous batches) drawn from $\mathcal{S}$ at $\mathcal{T}_i$. Therefore, Theorem 1 can be extended for analyzing the case where the data distribution is shifting smoothly.

---

> ### Author Response · Authors · 2022-08-02
> **Author Response (Comment 6-11)**
>
>
> **Comment 7**. Can the authors comment on how the model deals with a case where knowledge previously seen is encountered again after initiating a new model?
>
> **Answer 7**. We provide the pseudocode of the ODDL in Appendix-E from Supplemental Material. Since the proposed dynamic expansion mechanism (Eq.(18)) compares the discrepancy between each previously learnt component and the currently updated memory buffer, the model does not expand the network architecture when seeing previously learnt knowledge because the left-hand-side of Eq.(18) would be very small. In addition, the proposed sample selection (Eq.(20) of the paper) would also filter out these similar samples from the memory buffer during the training. \\
>
> **Comment 8**. Given that the memory is erased when a new component is initiated, wouldn't this lead to some redundancy in the learnt models?
>
> **Answer 8**. When building a new component, the last updated component is frozen to preserve the knowledge of the memory buffer. It is not necessary to allow the newly initiated component to learn the overlapping samples again and therefore we erase the memory buffer when the proposed ODDL performs the expansion. In addition, since the sample selection and dynamic expansion mechanism are performed when the memory buffer is full, clearing the memory buffer can also accelerate the model's training.
>
> **Comment 9**. Would it be possible to add a phase to recombine the knowledge encoded in the different components of the model?
>
> **Answer 9**. Yes, this is an excellent idea. Two solutions can be adopted. One way is to train a classifier as the student model along with the proposed ODDL in which we transfer the knowledge from all components to the student model using Knowledge Distillation (KD). The second approach is to search the statistically overlapping components and combine them using the KD. However, these two solutions would inevitable deteriorate the performance of ODDL because KD can not transfer all previously learnt information to a single model.
>
> **Comment 10**. Can the authors comment of the computational cost of their method, and how the model grows over time, as compared to the prior works?
>
> **Answer 10**.  When the memory buffer is not full, the computational costs of ODDL are about the training of the current expert, which is not high. In contrast, when the memory buffer is full, the sample selection and dynamic expansion evaluation are performed in each training step, leading to more computational costs. In addition, increasing the number of components in the ODDL would also lead to computational costs for the sample selection and dynamic expansion evaluation. Therefore, as learning more components over time, the model would increase the computational costs.
>
> CNDPM is another dynamic expansion model for TFCL, which dynamically builds new components through the Dirichlet process. During the training, each incoming sample is evaluated by every component in order to decide either the component selection for training or storing the incoming sample in the memory buffer. Therefore, learning more components over time, CNDPM would lead to growing computational costs. Compared with CNDPM, ODDL would require a bit more computational costs than CNDPM since the sample selection in the memory buffer of ODDL requires evaluating the discrepancy score for each memorized sample. However, we can accelerate the proposed sample selection and dynamic expansion evaluation by recording the discrepancy score for the memorized samples and only calculating the discrepancy score for each incoming sample once.
>
>
> **Comment 11**. The field has long focused on catastrophic forgetting. The analysis in this paper is also built around fighting this behavior. When training on a stream of data, however, one could argue that the present and future performance should have a higher weight. What do the authors think about this?
>
> **Answer 11**. The present and future performance can be seen as the Stability-Plasticity Dilemma in [1], [2], where 'Stability' denotes the ability of preserving previous knowledge and 'Plasticity' denotes the ability to adapt to a new task. However, the Stability and Plasticity in the model are hardly satisfied simultaneously because the model's weight continually changes over time. In addition, if the past and future samples of a data stream are sufficiently different, a single model would suffer from the interference between the present and future performance. We believe that the proposed ODDL can solve the Stability-Plasticity Dilemma well since it preserves previously learnt knowledge in the frozen components while building a new component to deal with the distribution shift.

---

> ### Author Response · Authors · 2022-08-02
> **Author Response (Comment 12)**
>
>
> **Comment 12**. Can their analysis and proposed method be adapted to look into forward transfer rather than preventing the model from forgetting?
>
> **Answer 12**. The proposed theory is originally designed for the forgetting analysis, but we can extend it to the forward transfer analysis by considering the following rationale. Based on Theorem 1 of the paper, we can derive a GB for a model at ${\mathcal{T}}_i$:
>
> $${\mathcal{R}} \big( h,{\mathcal{P}}_{i:i+1} \big) \le {\mathcal{R}} (h, {h}\_{{\mathcal{M}}\_i}, {\mathbb{P}}\_{{\mathcal{M}\_i}} \big) ) + {\mathcal{L}}\_{\widehat{d}} \big( {\mathcal{P}}^{\mathcal{X}}\_{i:i+1},{\mathbb{P}}^{\mathcal{X}}\_{{\mathcal{M}}\_i} \big) + \eta \big( {\mathcal{P}}\_{i:i+1},{\mathbb{P}}\_{{\mathcal{M}}\_i} \big)
> $$  (Eq.(1))
>
> where ${\mathcal{P}}\_{i:i+1}$ denotes the probabilistic representation of the new incoming data batch $\{ {\bf X}^b\_{i+1}, {\bf Y}^b\_{i+1} \}$ from the data stream. Eq.(1) describes the performance on the new data batch, achieved by a model trained on the memory buffer at the previous training step (${\mathcal{T}}\_i$). It observes that if the new data batch shares similar knowledge with the memory buffer, RHS of Eq.(1) would be decreased, leading to improved performance on the new data batch. Therefore, Eq.(1) can evaluate the forward transfer ability of a model on the incoming data batch at each training step.
>
> We believe the proposed model would have forward transfer ability by considering three approaches:
>
> 1. We can initialize a new component by using the weights from an existing component that shares similar knowledge with the incoming data batch.
>
> 2. We can adopt meta-learning to initialize a new component to adapt to incoming samples quickly.
>
> 3. We can reuse some previously learnt and frozen network parameters when learning incoming samples, which would have a positive forward transfer.
>
> ----
>
> ## Reference
>
> - [1] Gail A Carpenter and Stephen Grossberg. Art 2: Self-organization of stable category recognition codes for analog input patterns. {\em Applied Optics}, 26(23):4919–4930, 1987.
>
> - [2] Martial Mermillod, Aurélia Bugaiska, and Patrick Bonin. The stability-plasticity dilemma: Investigating the continuum from catastrophic forgetting to age-limited learning effects. {\em Frontiers in Psychology}, 4:504, 2013

---

> ### Comment · Reviewer_aR21 · 2022-08-08
> **Thanks for the answers**
>
> I thank the authors for the detailed answers.
>
> After reading the other reviews and the authors answers, I still think that the paper provides a solid contribution, and therefore keep my score.

---

### Meta-Review · Area_Chair_Txrt · 2022-08-25

**Recommendation:** Accept
**Confidence:** Certain

**Metareview:**

This paper is well-written. All reviewers agree the paper has solid contribution for task-free continual learning by providing theoretical foundation first, and the developing methods inspired by their theory. Empirical studies by comparing with a number of baseline methods demonstrate the effectiveness of the proposed method. All reviewers' concerns during the discussion phase have been addressed. Hence, I recommend acceptance.

**Award:**

No

---

### Decision · Program_Chairs · 2022-09-14

Accept